# Rapid processing of neutral and angry expressions within ongoing facial stimulus streams: Is it all about isolated facial features?

Antonio Schettino[1,2]*, Emanuele Porcu[3], Christopher Gundlach[4], Christian Keitel[5,6�], Matthias M. Müller[4�]

**1** Institute for Globally Distributed Open Research and Education (IGDORE), Ubud, Indonesia, **2** Erasmus Research Services, Erasmus University Rotterdam, Rotterdam, The Netherlands, **3** Institut für Psychologie II, Otto-von-Guericke-Universität Magdeburg, Magdeburg, Germany, **4** Institut für Psychologie, Universität Leipzig, Leipzig, Germany, **5** Psychology, University of Stirling, Stirling, United Kingdom, **6** Institute of Neuroscience and Psychology, University of Glasgow, Glasgow, United Kingdom

उ These authors contributed equally to this work.
\* antonio.schettino@igdore.org

**Data Availability Statement:** Raw and pre-processed data, materials, and analysis scripts are available at https://osf.io/uhczc/.

## Abstract

Our visual system extracts the emotional meaning of human facial expressions rapidly and automatically. Novel paradigms using fast periodic stimulations have provided insights into the electrophysiological processes underlying emotional content extraction: the regular occurrence of specific identities and/or emotional expressions alone can drive diagnostic brain responses. Consistent with a processing advantage for social cues of threat, we expected angry facial expressions to drive larger responses than neutral expressions. In a series of four EEG experiments, we studied the potential boundary conditions of such an effect: (i) we piloted emotional cue extraction using 9 facial identities and a fast presentation rate of 15 Hz (N = 16); (ii) we reduced the facial identities from 9 to 2, to assess whether (low or high) variability across emotional expressions would modulate brain responses (N = 16); (iii) we slowed the presentation rate from 15 Hz to 6 Hz (N = 31), the optimal presentation rate for facial feature extraction; (iv) we tested whether passive viewing instead of a concurrent task at fixation would play a role (N = 30). We consistently observed neural responses reflecting the rate of regularly presented emotional expressions (5 Hz and 2 Hz at presentation rates of 15 Hz and 6 Hz, respectively). Intriguingly, neutral expressions consistently produced stronger responses than angry expressions, contrary to the predicted processing advantage for threat-related stimuli. Our findings highlight the influence of physical differences across facial identities and emotional expressions.

## Introduction

The human brain is capable of rapidly processing differences in facial expressions and identifying those that signal threat, presumably due to the survival advantage of such an ability [1,2]. From an evolutionary point of view, speed of processing, identification, and discrimination

**Funding:** This work was supported by grants from the Deutsche Forschungsgemeinschaft to MMM (MU972/22-1, MU972/22-2). AS was also supported by Ghent University (BOF14/PDO/123). The funding sources had no involvement in the study design; collection, analysis, and interpretation of data; writing of the report; and decision to submit the article for publication.

**Competing interests:** The authors have declared that no competing interests exist.

are essential for the survival of an individual within a group, because not all members will exhibit identical (facial) expressions: in a threatening situation, some might express anger—i.e., aggression and the will to fight—, whereas others might express anxiety and a tendency for flight. This communication of contradictory information must be rapidly processed to allow the individual to adapt behavior accordingly.

An elegant way to investigate the time required for emotion detection and discrimination is to show stimuli in a rapid serial visual presentation stream (RSVP) at a certain frequency. In RSVP paradigms, each cycle is defined by the onset of a new image; thus, each image serves as a forward mask for the subsequent image and as a backward mask for the preceding one [3]. Such frequency-tagged stimulation streams evoke the steady-state visual evoked potential (SSVEP). SSVEPs have the same fundamental frequency as the RSVP stream and may include higher harmonics [4,5]. Attending to RSVP streams results in a modulation of the SSVEP, either quantified as an increase in amplitude [6–8] or enhanced inter-trial phase consistency [9,10].

A number of studies demonstrated that the presentation of emotionally arousing stimuli resulted in enhanced SSVEPs compared with neutral stimuli [e.g., 11–13]. Manipulating the presentation frequency of the RSVP allows to test the minimal presentation times for emotional cue extraction by analyzing SSVEPs in the frequency domain. In a recent study by Bekhtereva and Müller [14], we showed that presenting complex images from the International Affective Picture System [IAPS; 15] at a rate of 15 Hz (i.e., about 67 msec per image) was too rapid to enable emotional cue extraction, resulting in no discernible changes in SSVEPs between emotional and neutral images. Conversely, a 6 Hz presentation (i.e., about 167 msec per image) resulted in a significant modulation of SSVEPs as a function of emotional content. This presentation time was close to what was previously reported by Alonso-Prieto et al. [16] using RSVP paradigms that discriminated between faces with different identities. In that study, SSVEPs increased when different faces were presented in an RSVP compared with presenting the same face for the entire stream, a finding consistent with repetition suppression effects [17,18]. Alonso-Prieto et al. [16] reasoned that a longer interstimulus interval allowed for the full development of the N170, an event-related potential (ERP) component classically linked to the identification and discrimination of faces [19–21]. This, in turn, would enable sufficient time for individual face identification and emotional cue extraction. Subsequently, Liu-Shuang et al. [22] published an extension of the aforementioned stimulation approach by introducing regularities after a fixed number of cycles (i.e., images). In other words, a certain exemplar ("oddball") was presented after four filler faces, which resulted in an SSVEP at the base frequency (6 Hz) and a second peak in the frequency spectrum at a slower rate of the oddball presentation (1.2 Hz). Interestingly, the frequency spectrum lacked such a 1.2 Hz peak when faces were presented upside down, an experimental condition typically used to control for low-level visual features because inverted faces share the same first-order features (e.g., eyes, nose) but disrupt second-order configuration, i.e., the relation between features [e.g., 23].

Given that previous work mainly focused on face identification using RSVP protocols with either same/different faces [22] or faces among natural images [24], in 2014 we started a series of studies to extend RSVP stimulation using complex IAPS images and faces by including regular oddball stimuli in the RSVP. The present report is a comprehensive summary of the results we obtained in a research programme consisting of two pilot studies (N = 16/16) and two experiments with larger samples (N = 31/30), for a total of 93 recorded participants. Similar to our previous work using complex pictures [14,25–27], the purpose of this series of studies was to test whether—and under which conditions—the visual system is able to identify a regularly presented neutral or emotional face (angry in the present experiments) within an RSVP stream of different facial identities and expressions as "fillers". Given the alleged

motivational relevance of angry faces, we initially expected processing advantages—and, consequently, increased SSVEPs—for regularly presented angry as compared to neutral faces. By using different facial identities and expressions as "fillers", our experimental protocol extended the fast periodic oddball stimulation introduced by Liu-Shuang et al. [22] by using a different face and/or facial expression for each cycle (except the regularly presented items). In doing so, we significantly increased stimulus variability and set the visual system under a "perceptual stress test" [27] to probe boundary conditions of rapid facial emotional cue extraction. By presenting RSVP streams with inverted faces of exactly the same sequences, we additionally tested whether any observed SSVEP modulations may be due to low-level visual features and/or second-order relationships between facial elements. We initially explored much shorter presentation times—well above 6 Hz—to probe the temporal boundary conditions established for complex images in previous work [14]. Specifically, we tested whether the modulation of the oddball frequency via fixed exemplars requires the full processing of an individual face or, instead, regular presentations within a longer RSVP stream may allow for a significantly shorter presentation time because the gist of the emotional expression can be integrated through regular presentation. To that end, similar to our first IAPS study [14], we piloted with a stimulation frequency of 15 Hz (i.e., cycle length ~67 msec). To ensure that participants attended the RSVP, we included an orthogonal attention task instructing participants to detect and respond to a colored dot that was unpredictably overlayed on the face stream. We started our experimental series with the initial hypotheses that, if the emotional facial expression could be extracted reliably from the stream of faces, SSVEPs at the regularity frequency should be measurable for the regular but not the irregular conditions. SSVEPs for angry faces should be higher than for neutral images and should be also higher for upright than for inverted faces.

In the meantime, Dzhelyova and colleagues [28] published a study based on a similar rationale and methodology. These authors presented an RSVP at about 6 Hz with one neutral face of one individual and inserted, once every 5th stimulus, a face with an emotional expression of that individual, respectively. In a second experiment, they increased the base presentation rate of neutral faces to 12 Hz, with an emotional face presented every 9th stimulus. They analysed SSVEPs to the oddball frequency (about 1.2 Hz in Experiment 1 and about 1.3 Hz in Experiment 2) and additionally explored all higher harmonics, i.e, integer multiples of the oddball frequency. Identical to our protocol, they presented streams with inverted faces at the same stimulation frequencies, to control for the influence of low-level features. SSVEPs were found to be above noise level at the sum of all oddball frequencies (i.e., including all higher harmonics) elicited by emotional faces. Interestingly, this was also true for inverted faces, although the effect was smaller in magnitude compared to upright faces when looking at more occipito-temporal electrodes.

In their experimental paradigm, Dzhelyova et al. [28] always switched from a neutral to an emotional expression. This methodological choice might have overlooked systematic low-level, physical differences between stimuli, which could also explain the reported SSVEPs above noise level in the inverted condition. In our studies, we used different emotional facial expressions and the regular expression was either an angry or a neutral face. In contrast to Dzhelyova et al. [28] and other studies with similar paradigms, our use of a range of expressions and identities also changed the status of the "oddball" to yet another facial expression in the RSVP, with the only difference that it was presented regularly among irregular presentations of filler faces. Throughout this manuscript, we will therefore refer to it as the *regular* rather than the oddball expression and term its presentation rate *regularity rate* and the corresponding neural response, if present, *regularity-driven* SSVEP.

Given the increased diversity of emotional facial expressions compared with only neutral faces as filler stimuli, we were able to better randomize physical stimulus differences which

may arguably influence the neural response to the regular presentations. In addition, we swapped emotional expressions and either embedded angry or neutral faces regularly within the RSVP. This allowed us to test whether angry faces elicited a larger response compared to neutral expressions, theoretically expected under the assumption that threatening information is of greater behavioural significance and thus leads to prioritised neural processing. Furthermore, we included another important control condition: we presented the respective RSVP stream with different facial expressions without any regular repetitions. This important manipulation controls for an inherent problem of the stimulation protocol: the SSVEP driven by the regular presentation (i.e., the "oddball" in previous studies) is always a subharmonic of the RSVP frequency. Thus, only testing whether a regularity-driven SSVEP is above noise-level cannot exclude that it is, in fact, a subharmonic of the SSVEP to the RSVP rate and does not specifically indicate a functional processing of such regularity. With our control condition, presenting faces in irregular order only as a contrast, we were able to test for that possible confound.

## Pilot 1

### Materials and methods

**Participants.** Sixteen participants (10F/6M, median age 22.5 years, range 19–31, normal or corrected-to-normal vision, no self-reported history of neurological or psychiatric disorders) were recruited from the student population and the general public. Participants gave informed written consent prior to the experiment and were financially reimbursed €12 afterwards. All studies reported here were approved by the ethics committee of the University of Leipzig.

**Stimuli.** Stimuli were selected from *NimStim*, a validated database of facial expressions free for academic use [29]. This pilot experiment included identities #21, #22, #23, #25, #26, #33, #34, #36, and #37 (all males); the selected expressions were neutral, angry, and happy (all with closed mouth). All stimuli were resized to 152 x 190 pixels using Irfanview (https://www.irfanview.com/) and converted to grayscale in MATLAB R2015a (The MathWorks, Inc., Natick, MA) via the standard NTSC conversion formula used for calculating the effective luminance of a pixel: *intensity* = (0.2989 * *red*) + (0.5870 * *green*) + (0.1140 * *blue*) (see https://tinyurl.com/rgb2gray). Their luminance was matched using the *SHINE* toolbox [30]. To remove external facial features (e.g., hair and ears) and to standardize the spatial layout occupied by each face, stimuli were enclosed in an oval frame at presentation.

**Procedure.** After signing the informed consent, participants were seated comfortably in an acoustically dampened and electromagnetically shielded chamber and directed their gaze towards a central fixation cross (0.8˚ x 0.8˚ of visual angle) displayed on a 19-inch CRT screen (refresh rate: 60 frames per second; 640 x 480 pixel resolution) placed at a distance of 80 cm. The experimental stimulation consisted of a rapid serial visual presentation (RSVP) of face images showing each stimulus (size = 3.5˚) in the center of the screen. The RSVP was presented at a rate of 15 faces per second (15 Hz), resulting in a presentation cycle of four frames (cycle length ~67 msec). Each face was shown during the first 50% of the frames of one cycle, producing a 50/50 on/off luminance flicker. Within the RSVP in each trial, faces were randomly drawn and organized in triplets. Depending on the experimental condition, the first image of each triplet was either an angry or a neutral face. For position two and three, images were pseudo-randomly drawn from the remaining expression categories so that emotional expressions were evenly distributed. Faces within one triplet were not allowed to re-occur in the following trial, to avoid short-term repetitions of identical faces. Happy faces were never presented regularly and only served as filler items.

In addition to the physical RSVP frequency (*stimulation frequency*), this presentation proto-col introduced a second rhythm defined by the regular occurrence of faces of one emotional category with each third face. The category *angry* or *neutral* thus repeated at 5 Hz, i.e., at one-third of the RSVP rate (*regularity frequency*). We further added a third *irregular* condition, for which image sequences were created by randomly drawing from all emotional categories (i.e., no regularity at 5 Hz). As control conditions, we mirrored the set-up of upright-face RSVPs (regular angry, regular neutral, and irregular) but presented all stimuli upside-down, i.e. inverted (see *Fig 1*).

At the beginning of each trial, participants were presented with a fixation cross for 1.2 sec. Subsequently, the RSVP was presented for 3.8 sec. Participants were instructed to press the spacebar on a standard QWERTZ USB keyboard any time they detected a turquoise dot (RGB: 128, 128, 196; diameter = 0.3˚ of visual angle) briefly displayed within the face area (2 consecu-tive frames = 33 msec). Targets occurred in 40% of trials and up to three times in one trial with a minimum interval of 600 msec between onsets. At the end of each trial, the fixation cross remained on screen for an extra 1 sec, allowing participants to blink before the next trial started.

We presented a total of 576 trials (96 trials per condition), divided into 8 blocks (~6 min 20 sec each). Before the start of the experiment, participants performed a few blocks of training. After each training and experimental block, they received feedback on their performance (average hit rate, reaction times, and number of false alarms within the block).

To ensure that our pre-selected facial expressions were processed in accordance with nor-mative categorization, at the end of the main task we asked participants to judge the level of conveyed anger and happiness using a 9-point Likert scale (1: very low anger/happiness; 9 very high anger/happiness) (see results in the *Supplementary Materials*).

**EEG recording and preprocessing.** Electroencephalographic activity (EEG) was recorded with an ActiveTwo amplifier (BioSemi, Inc., The Netherlands) at a sampling rate of 256 Hz. Sixty-four Ag/AgCl electrodes were fitted into an elastic cap following the international 10/20 system [32]. Electrodes *T7* and *T8* of the standard BioSemi layout were moved in position *I1* and *I2* to increase spatial resolution at occipital sites. The common mode sense (*CMS*) active electrode and the driven right leg (*DRL*) passive electrode were used as reference and ground electrodes, respectively. Horizontal and vertical electrooculogram (*EOG*) were monitored using four facial bipolar electrodes placed on the outer canthi of each eye and in the inferior and superior areas of the left orbit.

EEG preprocessing was performed offline with custom MATLAB scripts and functions included in EEGLAB v14.1.1b [33] and FASTER v1.2.3b [34] toolboxes. The continuous EEG signal was referenced to the average activity of all electrodes. After subtracting the mean value of the waveform (DC offset), we assigned electrode coordinates and segmented the signal into epochs time-locked to the beginning of the flickering stimulation (0–3.8 sec). We discarded all trials with behavioral responses (N = 216), leaving 360 epochs (60 per condition). After re-referencing to electrode *Cz*, FASTER was used for artifact identification and rejection (see commented script at https://osf.io/au73y/) using the following settings: (i) over the whole nor-malized EEG signal, channels with variance, mean correlation, and Hurst exponent exceeding $z = \pm3$ were interpolated via a spherical spline procedure [35]; (ii) the mean across channels was computed for each epoch and, if amplitude range, variance, and channel deviation exceeded $z = \pm3$, the whole epoch was removed; (iii) within each epoch, channels with vari-ance, median gradient, amplitude range, and channel deviation exceeding $z = \pm3$ were interpo-lated; (iv) condition averages with amplitude range, variance, channel deviation, and maximum EOG value exceeding $z = \pm3$ were removed; (v) epochs containing more than 12 interpolated channels were discarded. We also discarded epochs whose signal exceeded 5

A

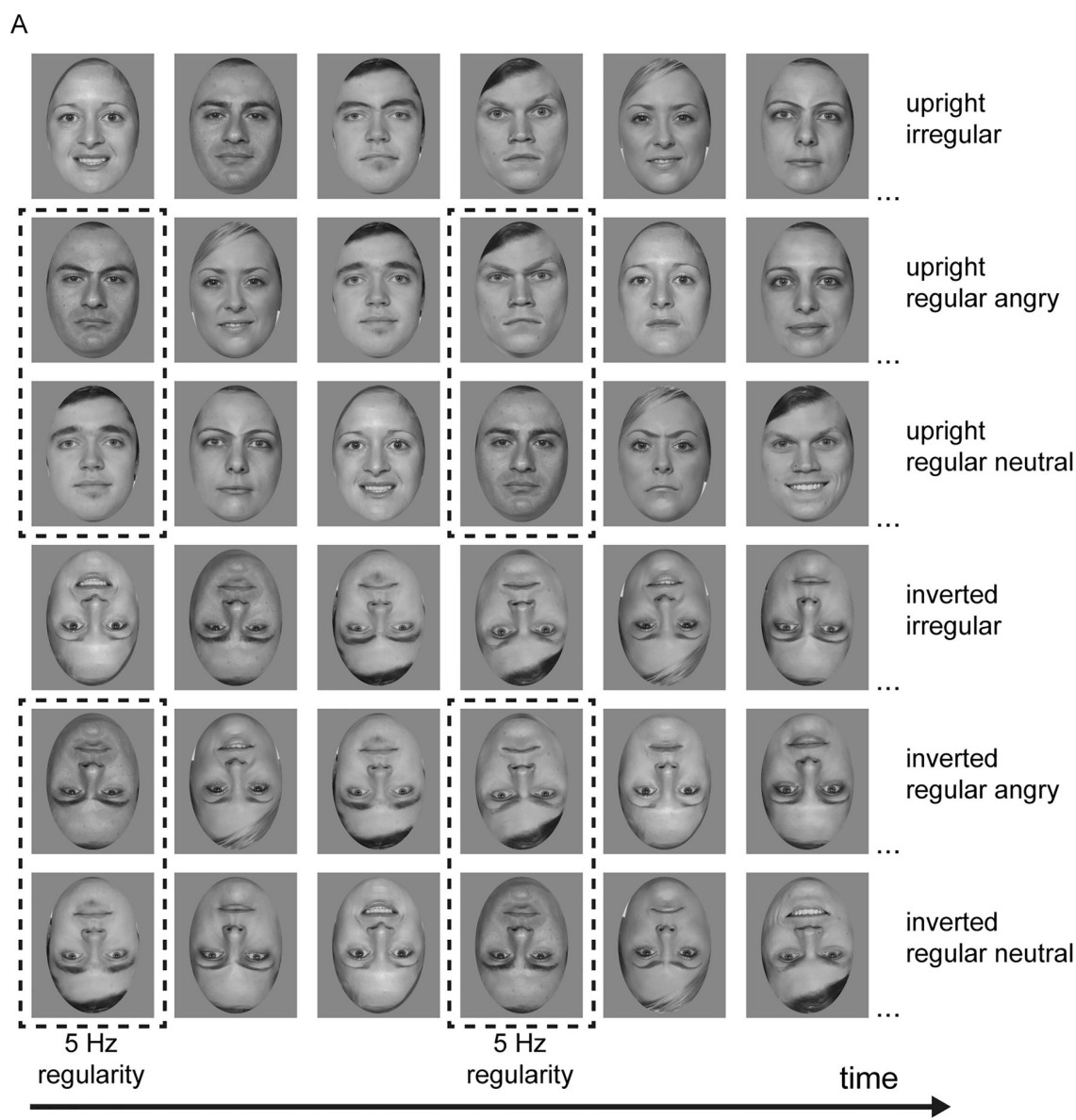

B

**Fig 1. Exemplar image sequences used in *Pilot 1*.** (**A**) Sequence of six exemplar images for each experimental condition. For the regular conditions, every third image contained a repetition of the same emotional category (angry or neutral). For the irregular condition, image triplets contained all three emotional categories (neutral, angry, happy) in random order. Images were presented either upright or upside down. Note that, for illustration purposes, we here show images from the Face Research Lab London Set [31], additionally modified in GIMP 2.10.14 (https://www.gimp.org/) to display anger. A set of different (only male) face identities was presented during the experiment, but cannot be shown here due to copyright restrictions. Please refer to https://www.macbrain.org/resources.htm for examples of faces used in the experiment. (**B**) Exemplar visual presentation of stimuli within each trial. Single images were presented for 33 msec followed by a fixation cross-only image for 33 msec, leading to a presentation rate of 15 Hz for the image stream with a regularity frequency of 5 Hz during the regular conditions. Note: image not to scale.

standard deviations from single- and all-channel mean kurtosis value, i.e., displaying abrupt spikes or flat activity. In addition, we verified that the spectral estimates would not deviate from baseline by +/-50 dB in the 0–2 Hz frequency window (indicating blinks) or +25/-100 dB in the 20–40 Hz frequency window (indicating muscular activity) [36]. The number of interpolated channels was low ($M$ = 4.06, $SD$ = 1.25). For an overview of the mean percentage of rejected epochs per condition, see *Table 1*. Finally, the resulting epoched data were re-referenced to the average activity of all scalp electrodes.

**Spectral decomposition of stimulus-driven EEG signals.** Artifact-free epochs were truncated to 3 sec starting 0.5 sec after RSVP onset, to exclude the initial event-related potentials. In truncated and detrended (i.e., linear trend removed) epochs, we quantified SSVEPs by means of Fourier transforms (technical details below) at each EEG sensor and for each condition, separately. We first inspected power spectra for peaks at RSVP (15 Hz) and face-regularity rates (5 Hz). Provided these peaks were present (which was the case in all participants), we used a recently developed spatial filtering approach to represent SSVEPs as an optimally weighted sum of all EEG sensors [37]. Spatial filters were derived separately for each experimental condition. We defined the signal bandwidth as +/-0.5 Hz, centered on the SSVEP frequency of the regularity-driven signal (5 Hz). Noise was defined as the spectral components centred on frequencies -/+ 1 Hz surrounding frequencies of interest with a bandwidth of +/-0.5 Hz (FWHM), respectively. To reduce numerical instabilities in spatial-filter estimation that may arise due to low trial numbers, the noise covariance matrix was regularised by adding 1% of the mean of its eigenvalues to its diagonal [38]. We opted to derive spatial filters per condition because topographical maps of regularity-driven SSVEPs differed substantially between conditions and thus prevented the alternative common-filter approach (one filter for all conditions) as well as the traditional definition of regions (i.e., electrode clusters with largest amplitude identified on topographical maps).

Note that we applied the same approach to conditions with irregular stimulus presentation (in the absence of a regularly repeated emotional expression) for reasons of consistency, although we did not expect to find a regularity-driven SSVEP. This can lead to overfitting noise and produce a spectral peak in the absence of an SSVEP [37]. However, in comparing regular and irregular conditions, the former should always produce a greater response when driving an SSVEP (see *Statistical Analysis* section below).

Filter-projected single-trial EEG time series were then multiplied with a Tukey window and subjected to Fourier transforms using the Fieldtrip function *ft_freqanalysis* (method 'mtmfft') [39]. To this end, data were zero-padded to a length of 10 sec, allowing for a unified frequency resolution of 0.1 Hz across experiments. From complex spectral representations of single trials we then computed the Cosine Similarity Index [40]—a measure of inter-trial phase clustering that can be interpreted similarly to the classic inter-trial coherence, ITC [41,42], or its derivative ITCz [43,44] but is less sensitive to the number of trials $n$—according to:

$$CS = \frac{2}{n(n-1)} \sum_{i=1}^{n-1} \sum_{j=i+1}^{n} cos\left(\theta_i - \theta_j\right) \tag{1}$$

where *cos* denotes the cosine transform and θ is the angle of the complex Fourier coefficient at the frequency of interest in separate trials *i* and *j*. Essentially, CS is based on quantifying and then summing the cosine of the angle differences between any two pairs of trials.

An overview of the CS values for each condition can be found in *Table 2*.

**Table 1. Percentage of rejected epochs after preprocessing.**

| experiment | variability | orientation | regularity | mean | st.dev. | min. | max. |
|---|---|---|---|---|---|---|---|
| Pilot 1 | | upright | angry | 13.85 | 3.57 | 8.33 | 21.67 |
| | | | neutral | 14.27 | 4.45 | 10.00 | 23.33 |
| | | | irregular | 16.35 | 4.69 | 10.00 | 25.00 |
| | | inverted | angry | 15.31 | 5.78 | 6.67 | 28.33 |
| | | | neutral | 15.73 | 4.64 | 6.67 | 26.67 |
| | | | irregular | 15.42 | 5.64 | 6.67 | 23.33 |
| Pilot 2 | high | upright | angry | 11.25 | 4.06 | 3.33 | 16.67 |
| | | | neutral | 15.62 | 5.36 | 6.67 | 23.33 |
| | | | irregular | 15.42 | 5.76 | 6.67 | 30.00 |
| | | inverted | angry | 13.54 | 6.18 | 3.33 | 26.67 |
| | | | neutral | 12.29 | 4.37 | 3.33 | 20.00 |
| | | | irregular | 15.62 | 6.64 | 3.33 | 30.00 |
| | low | upright | angry | 14.58 | 5.51 | 10.00 | 30.00 |
| | | | neutral | 13.75 | 5.64 | 3.33 | 26.67 |
| | | | irregular | 13.79 | 4.57 | 3.33 | 23.33 |
| | | inverted | angry | 18.12 | 8.08 | 3.33 | 33.33 |
| | | | neutral | 15.83 | 7.41 | 3.33 | 26.67 |
| | | | irregular | 13.54 | 6.29 | 3.33 | 26.67 |
| Experiment 1 | high | upright | angry | 15.73 | 6.87 | 3.45 | 33.33 |
| | | | neutral | 16.13 | 7.90 | 0.00 | 36.67 |
| | | | irregular | 18.45 | 8.05 | 3.33 | 40.00 |
| | | inverted | angry | 17.26 | 8.39 | 0.00 | 40.00 |
| | | | neutral | 16.44 | 8.02 | 3.33 | 36.67 |
| | | | irregular | 15.12 | 6.74 | 3.33 | 33.33 |
| | low | upright | angry | 19.78 | 8.80 | 3.33 | 43.33 |
| | | | neutral | 15.63 | 7.49 | 3.33 | 36.67 |
| | | | irregular | 17.15 | 9.16 | 3.33 | 36.67 |
| | | inverted | angry | 16.00 | 8.05 | 0.00 | 40.00 |
| | | | neutral | 15.96 | 6.04 | 3.33 | 30.00 |
| | | | irregular | 15.77 | 7.49 | 0.00 | 30.00 |
| Experiment 2 | high | upright | angry | 14.11 | 8.20 | 3.33 | 33.33 |
| | | | neutral | 15.37 | 7.24 | 3.33 | 36.67 |
| | | | irregular | 12.20 | 5.78 | 0.00 | 24.14 |
| | | inverted | angry | 14.13 | 6.19 | 0.00 | 30.00 |
| | | | neutral | 13.56 | 7.84 | 0.00 | 30.00 |
| | | | irregular | 14.33 | 6.62 | 3.33 | 36.67 |
| | low | upright | angry | 15.00 | 6.25 | 3.33 | 26.67 |
| | | | neutral | 12.81 | 7.07 | 0.00 | 23.33 |
| | | | irregular | 15.57 | 9.56 | 3.33 | 43.33 |
| | | inverted | angry | 13.89 | 6.33 | 0.00 | 26.67 |
| | | | neutral | 14.56 | 9.08 | 3.33 | 43.33 |
| | | | irregular | 15.02 | 9.01 | 3.33 | 40.00 |

Descriptive statistics of the percentage of removed trials after preprocessing, separately for all experiments and experimental conditions.

**Table 2. Cosine Similarity Index (CS), regularity frequency.**

| experiment | variability | orientation | regularity | CS |
|---|---|---|---|---|
| | | | | mean [95% CI] |
| Pilot 1 | | upright | angry | 0.03 [0.01, 0.04] |
| | | | neutral | 0.05 [0.04, 0.06] |
| | | | irregular | 0.00 [0.00, 0.01] |
| | | inverted | angry | 0.03 [0.01, 0.04] |
| | | | neutral | 0.05 [0.04, 0.06] |
| | | | irregular | 0.00 [-0.01, 0.01] |
| Pilot 2 | high | upright | angry | 0.06 [0.02, 0.09] |
| | | | neutral | 0.09 [0.06, 0.12] |
| | | | irregular | -0.02 [-0.04, -0.01] |
| | | inverted | angry | 0.03 [0.02, 0.04] |
| | | | neutral | 0.09 [0.06, 0.12] |
| | | | irregular | -0.01 [-0.02, 0.00] |
| | low | upright | angry | 0.02 [0.00, 0.04] |
| | | | neutral | 0.04 [0.03, 0.05] |
| | | | irregular | -0.02 [-0.03, -0.01] |
| | | inverted | angry | 0.03 [0.01, 0.05] |
| | | | neutral | 0.03 [0.01, 0.05] |
| | | | irregular | -0.03 [-0.04, -0.01] |
| Experiment 1 | high | upright | angry | 0.04 [0.03, 0.05] |
| | | | neutral | 0.06 [0.04, 0.07] |
| | | | irregular | 0.01 [0.00, 0.02] |
| | | inverted | angry | 0.03 [0.02, 0.04] |
| | | | neutral | 0.04 [0.03, 0.05] |
| | | | irregular | 0.00 [0.00, 0.01] |
| | low | upright | angry | 0.03 [0.02, 0.04] |
| | | | neutral | 0.04 [0.02, 0.05] |
| | | | irregular | 0.01 [0.00, 0.02] |
| | | inverted | angry | 0.02 [0.01, 0.03] |
| | | | neutral | 0.03 [0.02, 0.04] |
| | | | irregular | 0.00 [0.00, 0.01] |
| Experiment 2 | high | upright | angry | 0.04 [0.03, 0.05] |
| | | | neutral | 0.06 [0.05, 0.08] |
| | | | irregular | 0.00 [0.00, 0.01] |
| | | inverted | angry | 0.04 [0.02, 0.05] |
| | | | neutral | 0.05 [0.04, 0.06] |
| | | | irregular | 0.01 [0.00, 0.01] |
| | low | upright | angry | 0.02 [0.01, 0.03] |
| | | | neutral | 0.03 [0.02, 0.04] |
| | | | irregular | 0.01 [0.00, 0.02] |
| | | inverted | angry | 0.02 [0.01, 0.02] |
| | | | neutral | 0.02 [0.01, 0.03] |
| | | | irregular | 0.00 [0.00, 0.01] |

Statistics of Cosine Similarity Index (CS) of the signals at the regularity frequency, separately for the different experiments and experimental conditions. Regularity frequencies: 5 Hz in Pilot 1 and Pilot 2, 2 Hz in Experiment 1 and Experiment 2.

**Statistical analysis.** CS values were analyzed with Bayesian multilevel regressions using *brms* [45], a user-friendly *R* package that interfaces with the probabilistic programming language *STAN* [46] to estimate posterior distributions of the parameters of interest via Markov Chain Monte Carlo (*MCMC*) algorithms [47]. All models were fitted using weakly informative priors, i.e., *Normal(0,3)* on beta coefficients and *Student(3,0,2)* on the standard deviation of varying effects (i.e., participants). As a response distribution function, a Gaussian distribution was chosen. Parameters were estimated using 8 MCMC chains with 8,000 iterations each, 4,000 warmup samples—to get the sequences closer to the mass of the posterior distributions and then discarded—, and a thinning interval of 2, to minimize sample autocorrelation. Thus, the total number of retained posterior samples per parameter was 16,000.

We verified model convergence by visually inspecting trace plots and graphical posterior predictive checks [48]. We also examined: (i) the ratio of effective numbers of samples—i.e., the effective number of samples divided by the total number of samples—, which we aimed to keep larger than 0.1 to avoid excessive dependency between samples; (ii) the Gelman-Rubin $\hat{R}$ statistic [49]—comparing the between-chains variability (how much do chains differ from each other?) to the within-chain variability (how widely did a chain explore the parameter space?) [50]—which, as a rule of thumb, should not be larger than 1.05 or chains may not have successfully converged; (iii) the Monte Carlo standard error (MCSE)—the standard deviation of the chains divided by their effective sample size—a measure of sampling noise [51].

Differences between conditions were assessed by computing the mean and the 95% highest density interval (HDI) of the difference between posterior distributions of the relevant parameters [51] and calculating evidence ratios (ERs), i.e., the ratios between the proportion of posterior samples on each side of zero. ERs can be interpreted as the probability of a directional hypothesis (e.g., "condition A is larger than condition B") against its alternative (e.g., "condition B is larger than condition A"). As a rule of thumb, we interpreted our results as providing "inconclusive" evidence for a specified directional hypothesis when 1 < ER < 3, "anecdotal" evidence when 3 < ER < 10, and "strong" evidence when ER > 10. When ER = *Inf*, the posterior distribution was completely on one side of zero, thus providing "very strong" evidence. Please note that contrasts between different conditions in the irregular stimulus presentations are excluded from the results because uninterpretable, due to the overfitting issue with the spatial filtering approach mentioned above (see section *Spectral decomposition of stimulus-driven EEG signals*).

Throughout the main text we report the results of the analysis carried out on the cosine-similarity values at the Fourier coefficients that correspond to the regularity frequency in respective stimulation conditions (also for irregular presentation conditions). The results of behavioral performance, post-experiment emotion ratings, and SSVEP activity at the stimulation frequency are described in their respective sections in the *Supplementary Materials*.

**Software.** Data visualization and statistical analyses were carried out in *R* v3.6.1 [52] via *RStudio* v1.2.1335 [53]. We used the following packages (and their respective dependencies):

- data manipulation: *tidyverse* v1.2.1 [54], *Rmisc* v1.5 [55];

- statistical analyses: *brms* v2.10 [56], *rstan* v2.19.2 [57];

  v

- isualization: *ggplot2* v3.2.1 [58], *ggpirate* v0.1.1 [59], *bayesplot* v1.7.0 [48], *tidybayes* v1.1.0 [60], *bayestestR* v0.3.0 [61], *BEST* v0.5.1 [62], *viridis* v0.5.1 [63], *cowplot* v1.0.0 [64];

- report generation: *knitr* v1.25 [65].

## Results

Irrespective of face orientation, regular conditions elicited larger SSVEPs relative to irregular presentations (see *Table 3*), indicating that our stimulation protocol produced the intended regularity-driven SSVEPs. This is further demonstrated by the prominent parieto-occipital topographies of SSVEP maxima in the regular conditions that are absent in the irregular conditions (see *Fig 2*). We also observed that angry and neutral conditions elicited comparable SSVEPs when upright and inverted. Interestingly, neutral regular conditions showed larger SSVEPs compared to angry, both when upright [ER = 162.27] and inverted [ER = 25.02].

## Discussion

In this first pilot experiment (*Pilot 1*), we tested for enhanced SSVEPs driven by the regular presentation of angry over neutral faces at a rate of five similar emotional expressions per second (5 Hz) embedded in a stream of 15 facial identities per second (15 Hz). Based on previous studies [e.g., 25], we expected enhanced SSVEPs for regularly presented angry compared to neutral faces. Instead, our results showed the opposite pattern: neutral faces drove a more robust response. Further, upright and inverted regular faces showed comparable SSVEPs, in contrast with an expected effect of face inversion [i.e., smaller SSVEPs for inverted faces; see

**Table 3. Results statistical analyses Cosine Similarity Index (CS) at regularity frequency.**

| experiment | variability | orientation | regularity | comparison | mean [95% HDI] | evidence ratio |
|---|---|---|---|---|---|---|
| Pilot 1 | | | angry | upright vs. inverted | 0.00 [-0.02, 0.02] | 1.38 |
| | | | neutral | | 0.01 [-0.01, 0.02] | 3.39 |
| | | upright | | neutral vs. angry | -0.03 [-0.05, -0.01] | **162.27** |
| | | | | irregular vs. angry | 0.02 [0.01, 0.04] | **234.29** |
| | | | | irregular vs. neutral | 0.05 [0.04, 0.06] | *Inf* |
| | | inverted | | neutral vs. angry | -0.02 [-0.04, 0.00] | **25.02** |
| | | | | irregular vs. angry | 0.03 [0.01, 0.04] | **1,141.86** |
| | | | | irregular vs. neutral | 0.05 [0.03, 0.06] | *Inf* |
| Pilot 2 | high | | angry | upright vs. inverted | 0.03 [-0.02, 0.07] | 8.91 |
| | | | neutral | | 0.00 [-0.06, 0.06] | 1.03 |
| | low | | angry | upright vs. inverted | -0.01 [-0.04, 0.02] | 2.54 |
| | | | neutral | | 0.01 [-0.01, 0.04] | 4.52 |
| | high | upright | | neutral vs. angry | -0.03 [-0.08, 0.02] | 9.09 |
| | | | | irregular vs. angry | 0.08 [0.04, 0.12] | **1,776.78** |
| | | | | irregular vs. neutral | 0.11 [0.07, 0.15] | *Inf* |
| | | inverted | | neutral vs. angry | -0.06 [-0.09, -0.02] | **614.38** |
| | | | | irregular vs. angry | 0.04 [0.02, 0.06] | **1,332.33** |
| | | | | irregular vs. neutral | 0.10 [0.06, 0.13] | *Inf* |
| | low | upright | | neutral vs. angry | -0.02 [-0.04, 0.01] | 8.19 |
| | | | | irregular vs. angry | 0.04 [0.02, 0.07] | **1,999.00** |
| | | | | irregular vs. neutral | 0.06 [0.04, 0.08] | *Inf* |
| | | inverted | | neutral vs. angry | 0.00 [-0.03, 0.04] | 1.29 |
| | | | | irregular vs. angry | 0.06 [0.03, 0.08] | **15,999.00** |
| | | | | irregular vs. neutral | 0.05 [0.03, 0.08] | **1,776.78** |
| | | upright | angry | high vs. low | 0.04 [-0.01, 0.08] | **12.69** |
| | | | neutral | | 0.05 [0.01, 0.09] | **90.95** |
| | | inverted | angry | high vs. low | 0.00 [-0.03, 0.03] | 1.09 |
| | | | neutral | | 0.06 [0.02, 0.10] | **409.26** |

(*Continued*)

**Table 3.** (Continued)

| experiment | variability | orientation | regularity | comparison | mean [95% HDI] | evidence ratio |
|---|---|---|---|---|---|---|
| Experiment 1 | high | | angry | upright vs. inverted | 0.01 [0.00, 0.02] | **17.16** |
| | | | neutral | | 0.01 [-0.01, 0.03] | 7.62 |
| | low | | angry | upright vs. inverted | 0.01 [0.00, 0.02] | **10.15** |
| | | | neutral | | 0.01 [-0.01, 0.02] | 2.63 |
| | high | upright | | neutral vs. angry | -0.02 [-0.04, 0.01] | **10.83** |
| | | | | irregular vs. angry | 0.03 [0.02, 0.04] | *Inf* |
| | | | | irregular vs. neutral | 0.05 [0.03, 0.07] | **15,999.00** |
| | | inverted | | neutral vs. angry | -0.01 [-0.03, 0.00] | **16.72** |
| | | | | irregular vs. angry | 0.03 [0.02, 0.04] | *Inf* |
| | | | | irregular vs. neutral | 0.04 [0.03, 0.05] | *Inf* |
| | low | upright | | neutral vs. angry | -0.01 [-0.02, 0.01] | 6.57 |
| | | | | irregular vs. angry | 0.02 [0.01, 0.03] | **245.15** |
| | | | | irregular vs. neutral | 0.03 [0.01, 0.05] | **1,141.86** |
| | | inverted | | neutral vs. angry | -0.01 [-0.02, 0.00] | **102.90** |
| | | | | irregular vs. angry | 0.01 [0.00, 0.02] | **409.26** |
| | | | | irregular vs. neutral | 0.03 [0.01, 0.04] | *Inf* |
| | | upright | angry | high vs. low | 0.01 [0.00, 0.03] | **30.37** |
| | | | neutral | | 0.02 [0.00, 0.04] | **39.51** |
| | | inverted | angry | high vs. low | 0.01 [0.00, 0.02] | **49.63** |
| | | | neutral | | 0.01 [0.00, 0.03] | **23.46** |
| Experiment 2 | high | | angry | upright vs. inverted | 0.00 [-0.01, 0.02] | 1.66 |
| | | | neutral | | 0.01 [-0.01, 0.03] | **11.44** |
| | low | | angry | upright vs. inverted | 0.00 [-0.01, 0.02] | 1.89 |
| | | | neutral | | 0.01 [0.00, 0.02] | **75.56** |
| | high | upright | | neutral vs. angry | -0.03 [-0.05, -0.01] | **107.11** |
| | | | | irregular vs. angry | 0.04 [0.02, 0.05] | *Inf* |
| | | | | irregular vs. neutral | 0.06 [0.05, 0.08] | *Inf* |
| | | inverted | | neutral vs. angry | -0.01 [-0.03, 0.00] | **17.14** |
| | | | | irregular vs. angry | 0.03 [0.01, 0.04] | **15,999.00** |
| | | | | irregular vs. neutral | 0.04 [0.03, 0.06] | *Inf* |
| | low | upright | | neutral vs. angry | -0.01 [-0.02, 0.00] | 9.51 |
| | | | | irregular vs. angry | 0.0 [-0.01, 0.02] | 3.35 |
| | | | | irregular vs. neutral | 0.02 [0.00, 0.03] | **189.48** |
| | | inverted | | neutral vs. angry | 0.00 [-0.01, 0.01] | 1.03 |
| | | | | irregular vs. angry | 0.01 [0.01, 0.02] | **940.18** |
| | | | | irregular vs. neutral | 0.01 [0.01, 0.02] | **2,284.71** |
| | | upright | angry | high vs. low | 0.02 [0.00, 0.04] | **42.24** |
| | | | neutral | | 0.04 [0.02, 0.05] | **2,284.71** |
| | | inverted | angry | high vs. low | 0.02 [0.01, 0.03] | **362.64** |
| | | | neutral | | 0.03 [0.02, 0.05] | **3,199.00** |

Statistical comparisons of Cosine Similarity values between all pairs of factor levels, separately for all experiments and experimental conditions. Mean and 95% HDI are related to differences in CS value between the respective comparison. Comparisons showing strong evidence against the hypothesis of no difference are presented in bold.

22]. We speculated that the high variability of the stimulus material—nine different facial identities displaying three emotional expressions to various degrees of intensity—might have been a confounding factor. Specifically, angry expressions may differ more between identities than

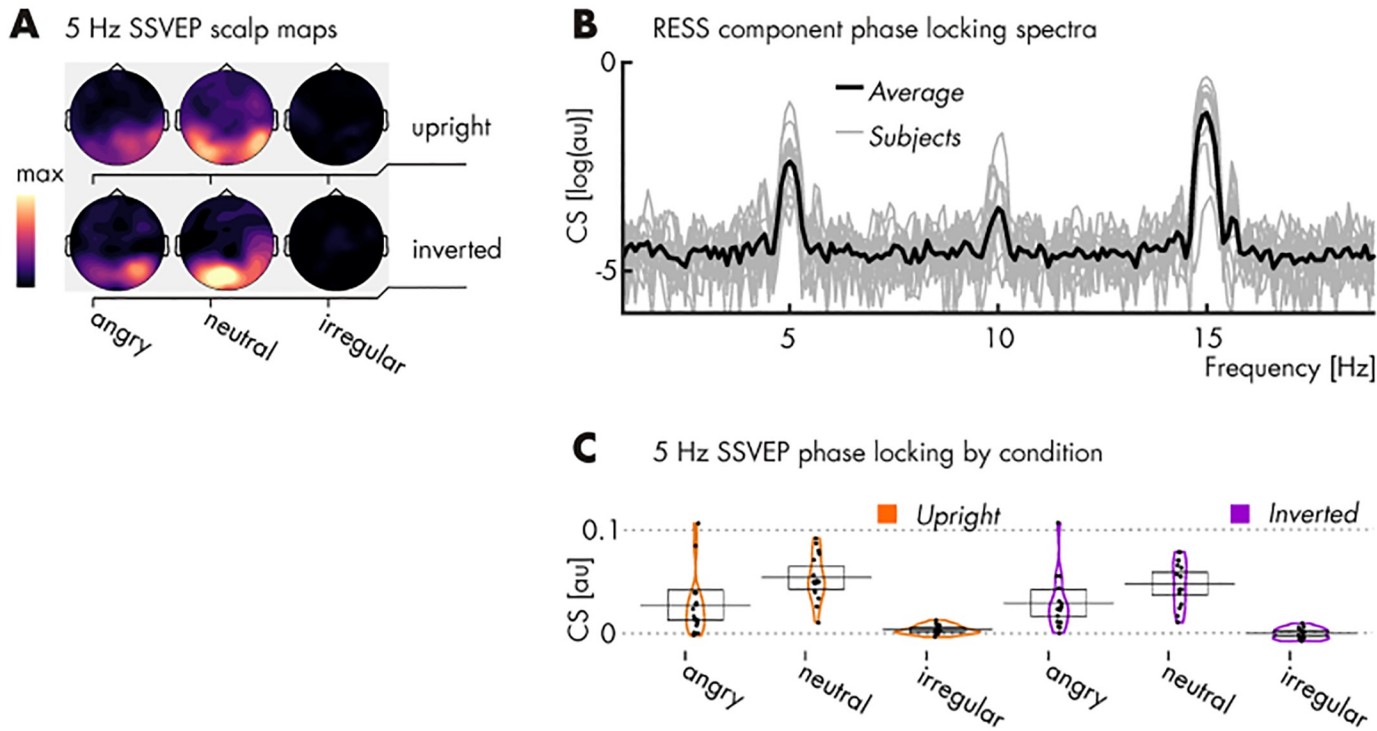

**Fig 2. Spectral characteristics of EEG responses to stimulation in _Pilot 1_.** (**A**) Topographical distributions of phase-locking, quantified as the cosine similarity (CS) index, at the regularity frequency of 5 Hz. Note the lack of phase-locking, i.e., consistent responses to the irregular stimulation conditions (maps use the same scale, in arbitrary units); (**B**) CS index of phase locking across the EEG spectrum (arbitrary scale) with the group average superimposed on single subject spectra, based on RESS spatial filter projections and collapsed across conditions featuring a regular presentation (i.e., excluding irregular stimulation conditions for their lack of signal; see panel **A**). For visualization only, CS has been converted to log(CS). (**C**) CS at the regularity frequency for each participant (single dots) and condition. Mean values are marked by horizontal black lines and 95% confidence intervals represented as transparent boxes. See _Table 3_ for specific results of statistical comparisons.

neutral expressions, and this greater dissimilarity may have led to a less consistent brain response.

To evaluate this hypothesis, in a follow-up pilot study we decided to use two instead of nine face identities. We chose a female identity with relatively low variability between angry and neutral expressions and one male identity with relatively high variability (see _Fig 3_). Trials were split evenly into only presenting either the female or the male identity. We expected that using fewer identities might facilitate the extraction of emotional information from the fast RSVP, thus leading to a face inversion effect (larger responses for upright relative to inverted faces). Furthermore, using one identity per trial should attenuate effects of variability in low-level visual features across facial expressions, thereby facilitating the actual extraction of emotional information leading to the expected gain effect for angry faces (i.e., enhanced SSVEP). Supporting this assumption, Vakli et al. [66] showed that the SSVEP adaptation effect to prolonged stimulation was invariant to changes in facial expression only when the same identity was presented.

## Pilot 2

### Materials and methods

**Participants.** Sixteen participants (12F/4M, median age 23.5 years, range 19–35) were recruited from the student population and the general public. Inclusion criteria, informed consent, reimbursement, and ethical statement are identical to _Pilot 1_.

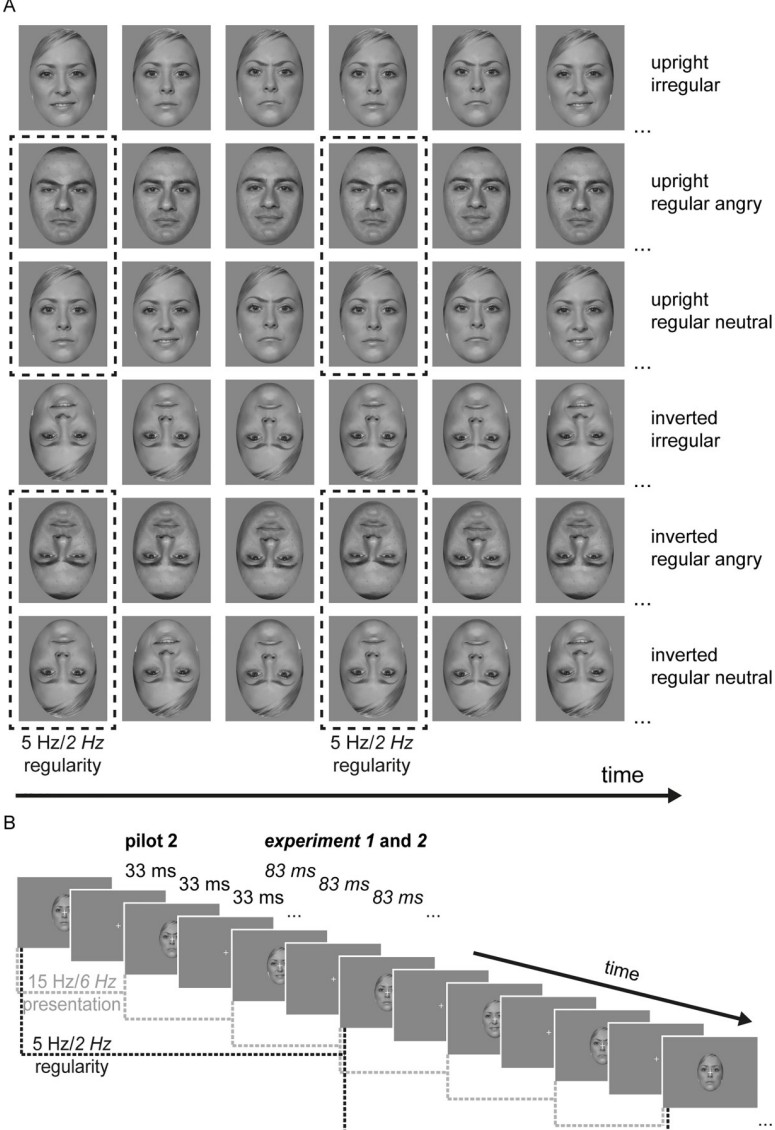

**Fig 3. Exemplar image sequences used in *Pilot 2* and *Experiment 1* and *2*.** (**A**) Sequence of six exemplar images for each experimental condition. For the regular conditions, every third image contained a repetition of the same emotional category (angry or neutral). For the irregular condition, image triplets contained all four emotional categories (neutral, angry, happy, disgusted) in random order. Images were presented either upright or upside down. For each trial only one of two face identities was presented, either with low (see top row) or high (see row two) dissimilarity between emotional expressions. Note that, for illustration purposes, we here show images from the Face Research Lab London Set [31], additionally modified in GIMP 2.10.14 (https://www.gimp.org/) to display anger. A set of different (only male) face identities was presented during the experiment, but cannot be shown here due to copyright restrictions. Please refer to https://www.macbrain.org/resources.htm for examples of faces used in the experiment. (**B**) Exemplar visual presentation of stimuli within each trial. Single images were presented for 33 msec in *Pilot 2* or 83 msec in *Experiment 1* and *2*, followed by a fixation cross-only image of similar duration, leading to a presentation rate of 15 Hz for the image stream in *Pilot 2* and 6 Hz in *Experiment 1* and *2*. The resulting regularity frequency was 5 Hz for *Pilot 2* and 2 Hz for *Experiment 1* and *2* during the regular conditions. Note: image not to scale.

**Stimuli.** From the NimStim database [29], we used identities #06 (female, low dissimilarity between emotional expressions) and #37 (male, high dissimilarity between emotional expressions). The selected expressions were neutral, angry, happy, and disgusted (all with closed mouth). Emotion dissimilarity was determined by calculating the structural similarity

index [following the procedure in ref. 67] between emotional expressions within each identity and selecting the ones with highest and lowest values. Picture resizing, luminance matching, and oval placement were identical to *Pilot 1*. See *Fig 3* for an illustration.

**Procedure.** The procedure was similar to *Pilot 1*, with the following exceptions. For each trial, only one of two identities was shown with varying emotional expressions. Both identities varied in the degree of similarity between emotional expressions, i.e., female identity with low dissimilarity and male identity with high dissimilarity. An additional emotional expression (disgusted, closed mouth) was presented to compensate for the reduced variation in picture content due to the presentation of only one identity per trial. Disgusted faces were only used as fillers and never presented regularly during the trials.

**EEG recording and preprocessing.** EEG recording and preprocessing were similar to the previous experiment, except that epochs displaying female and male identities (i.e., with low and high emotion dissimilarity) were kept separate. The number of interpolated channels was $M = 3.88$, $SD = 1.11$. See *Table 1* for the mean percentage of rejected epochs.

**Spectral decomposition of stimulus-driven EEG signals.** The spectral decomposition of the preprocessed EEG signal was identical to *Pilot 1*. Again, the stimulation frequency was 15 Hz, consequently setting the face regularity frequency to 5 Hz.

**Statistical analysis.** The statistical analyses were identical to *Pilot 1*. However, given the inclusion of another predictor (i.e., emotion dissimilarity), the combinations of our condition levels were 12: 2 (*face orientation*: upright, inverted) x 3 (*regularity*: neutral, angry, irregular) x 2 (*emotion dissimilarity*: low, high).

## Results

Similarly to *Pilot 1*, our data indicated that the regular presentation of emotional expressions drove SSVEPs at the regular rate of 5 Hz, whereas the irregular presentation did not (see *Fig 4* and *Table 3*).

Further mirroring the results of the previous pilot study, face inversion did not influence brain responses, with similar SSVEPs for upright and inverted regular faces (see *Table 3*).

Regarding contrasts between emotional expressions, regular neutral faces showed greater SSVEPs than angry faces only when inverted and when within-identity emotion variability was high [ER = 614.38].

This experimental design also allowed us to assess the impact of within-identity emotion variability on steady-state responses. Regular neutral faces of the highly variable (as compared with the less variable) identity elicited pronounced SSVEPs, both when upright [ER = 90.95] and inverted [ER = 409.26]. Angry, high variable faces elicited slightly enhanced SSVEPs only when upright [ER = 12.69].

## Discussion

We designed *Pilot 2* to test whether we would find the expected gain effect of emotional (angry) over neutral expressions while using only one facial identity per trial (instead of nine) at a presentation rate of 15 Hz and the regular occurrence of one emotional expression of 5 Hz. While the results of *Pilot 1* showed increased neural responses for regularly presented neutral faces, in *Pilot 2* this effect could not be observed. Neutral and angry faces elicited SSVEPs of comparable magnitude, with the notable exception of a persisting reversed effect (neutral > angry) for the high-variability identity when inverted. Possibly, angry facial expressions differed more strongly between identities in *Pilot 1*, thus driving less consistent brain responses. Using only one facial identity per trial presumably mitigated this confound [66]. Nevertheless, the hypothesized processing advantage for angry faces was still absent.

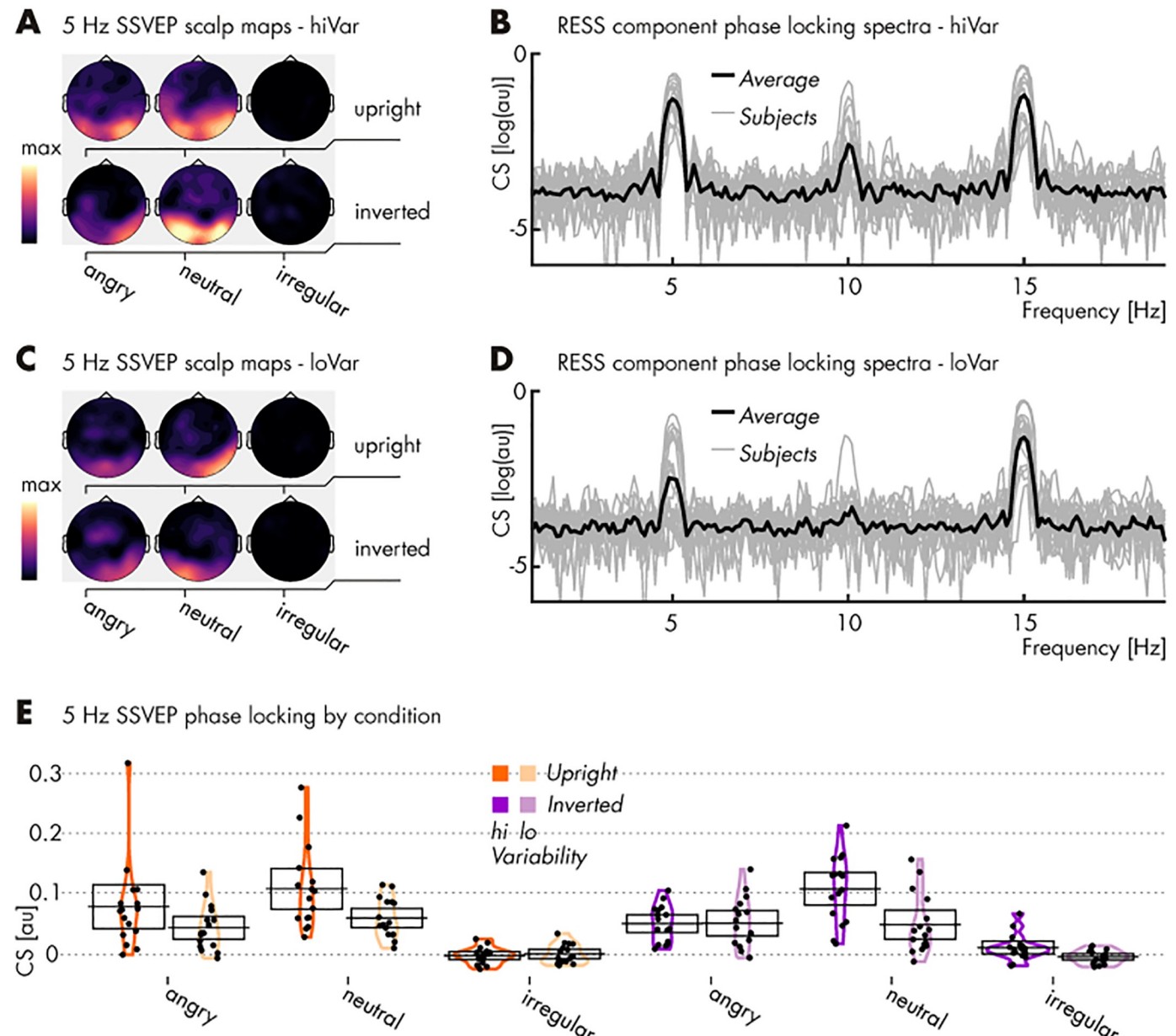

**Fig 4. Spectral characteristics of EEG responses to stimulation in *Pilot 2*.** (**A**) Topographical distributions of phase-locking, quantified as cosine similarity (CS), at the regularity frequency of 5 Hz for high variance stimuli. Note the lack of phase-locking, i.e. consistent responses to the irregular stimulation conditions (maps same scale, in arbitrary units); (**B**) CS index of phase locking across the EEG spectrum (arbitrary scale) with the group average superimposed on single subject spectra, based on RESS spatial filter projections and collapsed across conditions featuring a regular presentation of high variance stimuli (i.e., excluding irregular stimulation conditions for their lack of signal; see **A**). For visualization only, CS has been converted to log(CS); (**C** & **D**) Same as in panels **A** & **B** (identical scale) but for 5 Hz SSVEPs driven by low variance stimulation; (**E**) CS at the regularity frequency for each participant (single dots) and condition. Mean values are marked by horizontal black lines and 95% confidence intervals represented as transparent boxes. See *Table 3* for specific results of statistical comparisons.

Although our regularity frequency of 5 Hz seems close to the optimal frequency of 6 Hz for facial processing in an RSVP [e.g., 16], the actual RSVP was presented at 15 Hz which might have been too fast to extract the relevant emotional features while accentuating physical differences. Consistent with this potential caveat in our design, upright and inverted faces drove

5-Hz responses with comparable magnitudes [see also 28]. This absence of effect might indicate that the visual system was not properly able to extract the "facialness" of the stimuli, which typically produces enhanced neural responses for upright relative to inverted faces [23,68].

In the following *Experiment 1* we thus opted for an RSVP rate of 6 Hz, which decreased the regularity frequency to 2 Hz. We assumed that slowing down presentation times would provide the visual system with enough time and perceptual information to routinely process faces within the RSVP and consequently extract emotional expressions. By using the same two identities as in *Pilot 2* (thus controlling for low-level influences of facial variability), we expected to see the intuitive benefit of angry over neutral expressions as an increase in regularity-driven 2 Hz SSVEPs.

## Experiment 1

### Materials and methods

**Participants.** Thirty-one participants (22F/9M, median age 25 years, range 18–48) were recruited from the student population and the general public. This sample size was chosen based on available time and economic resources (no statistical *a priori* power analysis was conducted). Inclusion criteria, informed consent, reimbursement, and ethical statement are identical to the previous experiments.

**Stimuli.** Stimuli were identical to *Pilot 2* (see *Fig 3*).

**Procedure.** The procedure was similar to *Pilot 2* (see *Fig 3*), with the exception that the total presentation rate of faces was reduced to 6 per second (stimulation frequency = 6 Hz; cycle length ~167 msec) and regular faces occurred at a third of that rate (regularity frequency = 2 Hz). Due to the slower presentation rate, we extended the length of each trial to 7 sec (to ensure a number of cycles in the regularity frequency comparable with *Pilot 1* and *Pilot 2*) and increased the minimum idle interval between subsequent target events from 600 to 800 msec. Longer trials also required to subdivide the experiment into 16 experimental blocks, each consisting of 36 trials and lasting ~5 min.

**EEG recording and preprocessing.** EEG recording and preprocessing were identical to *Pilot 2*. The interpolated channels were $M = 3.26$, $SD = 1.37$. See *Table 1* for the mean percentage of rejected epochs.

**Spectral decomposition of stimulus-driven EEG signals.** The spectral decomposition of the preprocessed EEG signal was identical to the previous experiments except that the extended trials, now including a visual stimulation of 7 sec, where truncated to EEG data epochs with a length of 6.5 sec, starting 0.5 sec after RSVP onset. Spatial filters were centered on the new regularity frequency (2 Hz).

**Statistical analysis.** The statistical analyses were identical to *Pilot 2*.

### Results

Spectra in *Fig 5* demonstrate that slowing down the pace of the stimulation still elicited an RSVP-driven 6 Hz SSVEP and a regularity-driven 2 Hz SSVEP. Moreover, the 2 Hz SSVEP showed the same occipito-temporal topography as its 5 Hz counterpart in *Pilot 1* and *Pilot 2*. No 2 Hz SSVEP was driven in the irregular presentation condition.

Face orientation had a small influence on brain activity, with stronger SSVEPs during upright relative to inverted presentation of angry but not neutral faces (see *Table 3*). This effect was consistent for high- [ER = 17.16] and low-variability facial identities [ER = 10.15].

Regular neutral faces elicited stronger SSVEPs compared to angry faces in both orientations and emotion variabilities (*see Table 3*). When presented upright and variability was low, this effect was weaker [ER = 6.57].

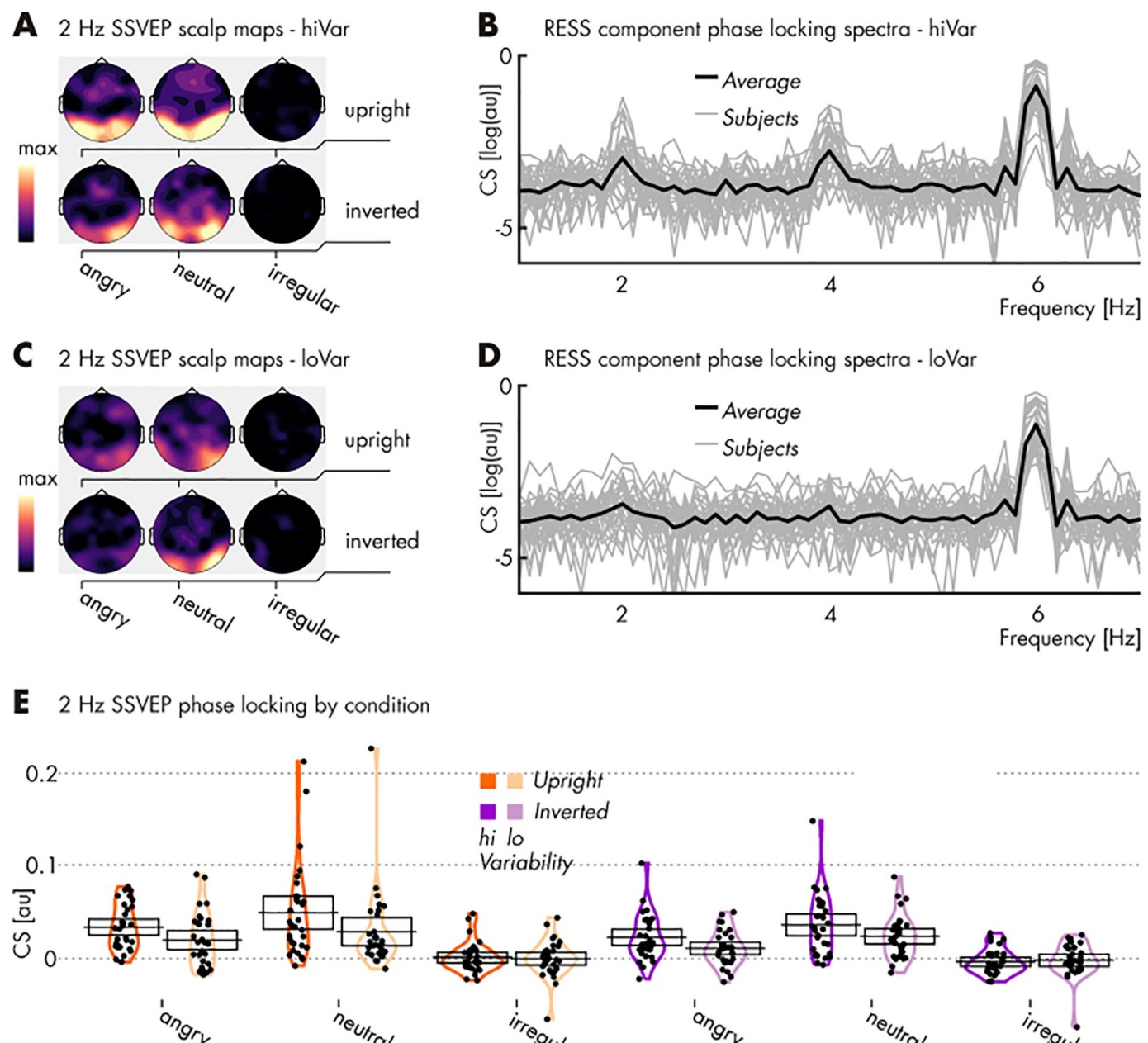

**Fig 5. Spectral characteristics of EEG responses to stimulation in *Experiment 1*.** (**A**) Topographical distributions of phase-locking, quantified as cosine similarity (CS), at the regularity frequency of 2 Hz for high variance stimuli. Note the lack of phase-locking, i.e. consistent responses to the irregular stimulation conditions (maps same scale, in arbitrary units); (**B**) CS index of phase locking across the EEG spectrum (arbitrary scale) with the group average superimposed on single subject spectra, based on RESS spatial filter projections and collapsed across conditions featuring a regular presentation of high variance stimuli (i.e., excluding irregular stimulation conditions for their lack of signal; see **A**). For visualization only, CS has been converted to log(CS); (**C & D**) Same as in panels **A** & **B** (identical scale) but for 2 Hz SSVEPs driven by low variance stimulation; (**E**) CS at the regularity frequency for each participant (single dots) and condition. Mean values are marked by horizontal black lines and 95% confidence intervals represented as transparent boxes. See *Table 3* for specific results of statistical comparisons.

Finally, when comparing high vs. low variability, regular angry and neutral faces of the highly variable identity elicited stronger SSVEPs in both orientations.

## Discussion

In *Experiment 1* we tested whether we could observe preferential attention allocation towards threatening faces using a slower presentation rate (6 Hz), with one facial identity per trial that displayed a particular emotion (angry or neutral) every third face (2 Hz). Results showed a difference in brain responses driven by the two orientations: upright faces drove a stronger response compared to their inverted counterparts, in line with the notion that naturally oriented faces experience a processing advantage in the visual system, likely due to their special relevance as a social cue display [23,68]. This effect also corroborated our choice of a slower optimal presentation frequency for faces, in line with earlier findings [14,16].

However, despite this progress in adjusting experimental parameters to reveal the hypothesized pattern of results, the expected processing advantage for emotional expressions was still not observed. In fact, the reversed effect—neutral expressions driving stronger responses, present in *Pilot 1* yet not observed in *Pilot 2*—was re-introduced in *Experiment 1* (with the exception of upright presented faces of the less variable variety). Additionally, note that the effect of variability (i.e., the physical difference between neutral and angry expressions of one facial identity) was highly consistent irrespective of the regular emotional expression or orientation of the stimuli. The latter effect indicated that physical differences could still be a confounding factor, also when comparing responses to emotional expressions [69].

We considered that *Pilot 1*, *Pilot 2*, and *Experiment 1* all required participants to perform a visual detection task that was superimposed on the RSVP of faces but completely unrelated. This manipulation may have diverted attentional resources necessary to process the stream of faces more comprehensively. Our recent experiment, in which participants were not performing a task but passively viewing an RSVP of IAPS scenes [27], showed stronger SSVEPs (indicative of enhanced processing) for emotional content at the expected regularity frequencies. In *Experiment 2* we therefore removed the dot detection task, changing the design to a passive viewing situation, and tested whether the findings observed with IAPS scenes would generalise to face stimuli.

## Experiment 2

### Materials and methods

**Participants.**    Thirty participants (27F/3M, median age 22 years, range 19–37) were recruited from the student population and the general public. Sample size rationale, inclusion criteria, informed consent, reimbursement, and ethical statement are identical to the previous experiments.

**Stimuli.**    Stimuli were identical to the previous experiment (see *Fig 3*).

**Procedure.**    The procedure was similar to *Experiment 1*. However, in contrast to the detection task used in the previous experiments, here we employed a simple passive viewing task: participants were asked to fixate the cross and attentively view the picture stream. We presented only 360 trials (60 trials per condition) because, in contrast to *Pilot 1*, *Pilot 2* and *Experiment 1*, here trials did not contain any targets and/or behavioral responses; therefore, all trials could be included in the analysis.

**EEG recording and preprocessing.**    EEG recording and preprocessing were similar to the previous experiment. The interpolated channels were $M = 3.50$, $SD = 1.09$. See *Table 1* for the mean percentage of rejected epochs.

**Spectral decomposition of stimulus-driven EEG signals.**    The spectral decomposition of the preprocessed EEG signal was identical to *Experiment 1*.

**Statistical analysis.**    The statistical analyses were identical to the previous experiment.

## Results

As in *Pilot 1*, *Pilot 2*, and *Experiment 1*, presenting one emotional expression regularly every three faces elicited an SSVEP at one third of the RSVP rate, i.e., 2 Hz. Note, however, that in this instance the 2-Hz SSVEP for low-variability angry upright faces was almost indiscernible from noise [ER = 3.35] (see *Table 3*). Regularity-driven SSVEP showed the same occipito-temporal topography and no 2 Hz SSVEP was driven in the irregular presentation condition (see *Fig 6*).

Face inversion modulated brain activity only for neutral expressions: irrespective of emotion variability, regular neutral conditions showed stronger SSVEPs when faces were upright (see *Table 3*). SSVEPs driven by regular angry faces were comparable for upright and inverted presentations. Regular neutral faces elicited larger SSVEPs than angry faces in both orientations, but only when variability was high [upright: ER = 107.11; inverted: ER = 17.14]. Finally, all conditions elicited greater SSVEPs in high compared to low within-identity emotion variability.

## Discussion

In *Experiment 2* we tested whether passive viewing changed SSVEPs during a 6Hz presentation rate (i.e., regular facial expressions at 2 Hz). Previous *Pilots* and *Experiment 1* had used a dot detection task, effectively withdrawing attention from the picture stream (i.e., faces were distractors). Nonetheless, redirecting the attentional focus did not produce the expected increase in brain responses for angry over neutral faces. Instead, and in accordance with the studies presented above, neutral faces drove stronger responses than angry faces, at least for the highly variable identity displaying greater dissimilarity between angry and neutral expressions.

It is worth pointing out that passive viewing produced a different face inversion effect compared to *Experiment 1*. Here, neutral but not angry faces consistently drove stronger responses when presented upright, whereas in *Experiment 1* we observed the opposite pattern of results. This might indicate that shifting the focus towards the face stream (instead of being a background distractor for an unrelated task) can alter the processing of the "facialness" of the stimuli [25]. Possibly, upright angry faces were stronger distractors in *Experiment 1*. Their threatening expressions may have increased the probability of them being recognized as human faces as opposed to other objects, presumably resulting in a stronger orientation effect. The passive viewing situation in *Experiment 2* perhaps counteracted this effect because attention was already fully allocated to the faces. Here, neutral faces might have been processed more intensely because the visual system expected emotional content given the RSVP context (i.e., mostly emotional expressions). The increased effort to extract emotional content from neutral faces might have given rise to the observed orientation effect during passive viewing. However, this speculative, post-hoc explanation must be evaluated against the fact that neutral faces generally drove stronger brain responses when compared with angry faces, a finding that aligns with other recent reports for non-facial emotional stimuli [26,70]. Furthermore, SSVEP amplitude modulation due to task demands were not directly compared, thereby limiting our conclusions.

Finally, also in *Experiment 2* the variability between emotional expressions of one facial identity was a strong contributor to the variance in the elicited brain response.

## General discussion

The experiments reported here were designed to investigate under which circumstances the human brain is able to extract emotional facial expressions embedded regularly in an RSVP stream. The RSVP elicited periodic brain responses (SSVEPs) at its presentation rate of 15 Hz

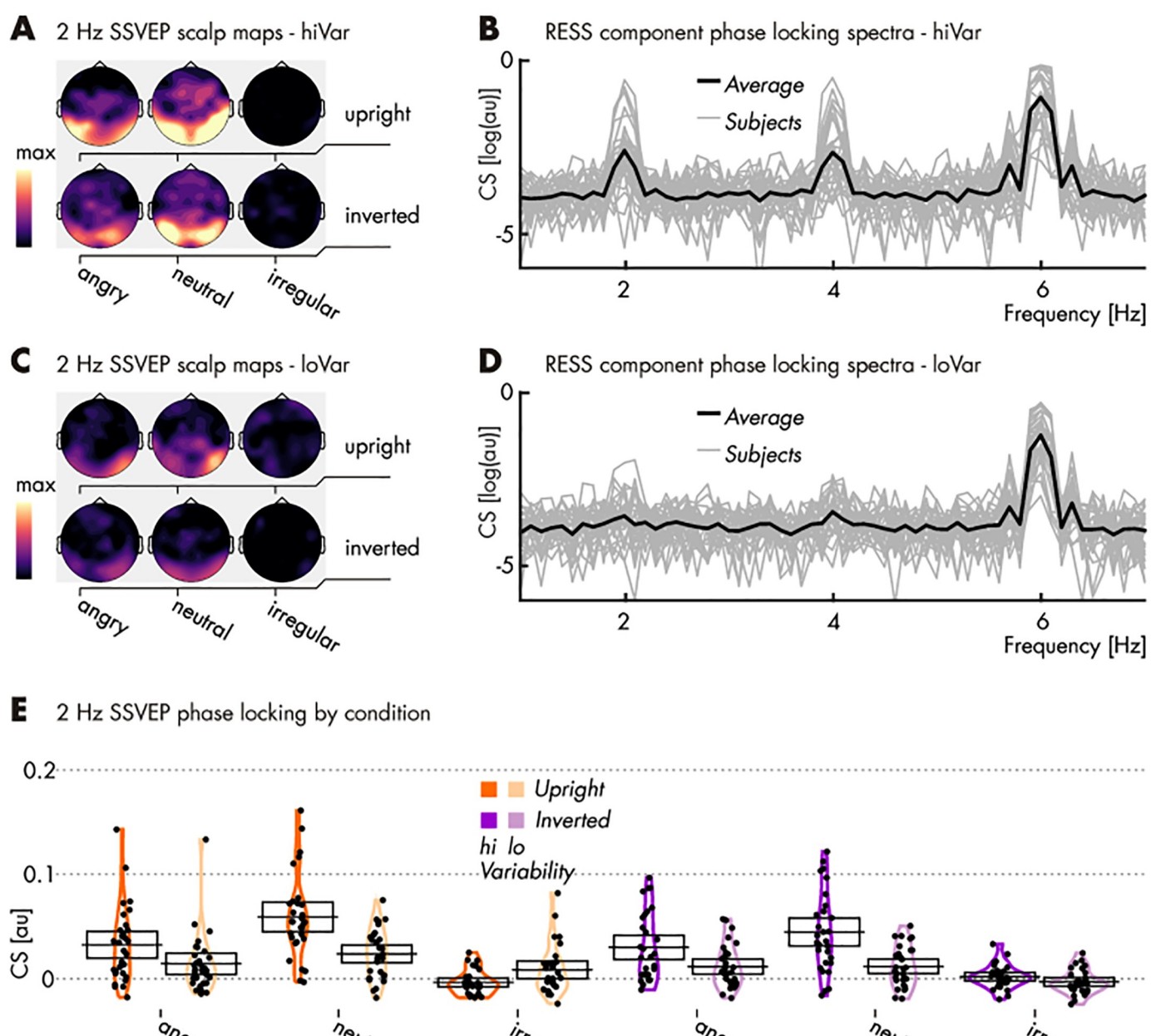

**Fig 6. Spectral characteristics of EEG responses to stimulation in *Experiment 2*.** (**A**) Topographical distributions of phase-locking, quantified as cosine similarity (CS), at the regularity frequency of 2 Hz for high variance stimuli. Note the lack of phase-locking, i.e. consistent responses to the irregular stimulation conditions (maps same scale, in arbitrary units); (**B**) CS index of phase locking across the EEG spectrum (arbitrary scale) with the group average superimposed on single subject spectra, based on RESS spatial filter projections and collapsed across conditions featuring a regular presentation of high variance stimuli (i.e., excluding irregular stimulation conditions for their lack of signal; see A). For visualization only, CS has been converted to log(CS); (**C** & **D**) Same as in panels **A** & **B** (identical scale) but for 2 Hz SSVEPs driven by low variance stimulation; (**E**) CS at the regularity frequency for each participant (single dots) and condition. Mean values are marked by horizontal black lines and 95% confidence intervals represented as transparent boxes. See *Table 3* for specific results of statistical comparisons.

in two *Pilot* studies and at 6 Hz in two *Experiments*. Based on previous studies that introduced regular oddballs in continuous RSVP streams [e.g., 16,22,26], we expected to observe an SSVEP response at the regularity frequency (one-third of the presentation rate; 5 Hz in the

*Pilots*, 2 Hz in the main *Experiments*). Further, if emotional expressions affords increased processing because motivationally salient, stronger regularity-driven brain responses ought to be expected for angry relative to neutral faces. We additionally implemented two control conditions: First, we presented RSVP streams with inverted faces, having identical physical properties compared with upright faces but resulting in delayed or disrupted identification processes [23,71,72] and expected diminished or absent electrophysiological responses at the regularity frequency [68]. Second, we presented RSVP streams consisting of the same face stimuli but without the regular repetition of emotional expressions, thereby assessing whether any observed responses at the regularity frequencies would indeed be related to processing the facial content or simply represent a subharmonic of the stimulation frequency.

Our analysis across all experiments revealed robust SSVEP responses at 15 Hz or 6 Hz, i.e. at the rate of respective RSVPs. In our *Pilots* (15 Hz RSVPs) we also observed a 5 Hz response for regularly presented stimuli, regardless of stimulus presentation (upright vs. inverted). For the two experiments with a 6 Hz RSVP, the 2 Hz SSVEPs were more prominent in streams with high within-expression variability compared to the streams with low variability. This was also observed with inverted faces, corroborating the results of our *Pilots*. In irregular streams, regularity-driven SSVEPs were absent, confirming that these brain responses cannot solely be ascribed to integer subharmonics of the driving frequencies of stimulus presentation but are indeed related to the processing of the regularly presented facial expression.

Regarding our main contrast of interest, we expected higher SSVEP responses for regular angry than for regular neutral expressions. This hypothesis stemmed from the results of a number of studies showing greater N170 amplitude for emotional compared to neutral faces [for a review, see ref. 73] as well as recent reports of enhanced SSVEPs for emotional oddball faces [e.g., 28]. Here, however, SSVEPs were enhanced for regularly presented neutral compared to angry faces in almost all our conditions, irrespective of face orientation. This effect cannot be explained by possible misinterpretations of the emotional cues within the faces: as outlined in the *Supplementary Material*, in all experiments angry faces were perceived as conveying more anger than neutral, happy, or disgusted faces. Therefore, it is unlikely that neutral faces were preferentially processed because they were less ambiguous compared to the other emotional expressions. Below we outline a few possible post-hoc interpretations of these results.

## Superposition of emotion-sensitive ERP components

What we initially considered counterintuitive findings could be re-interpreted under a different light when integrating recent published results. A study using IAPS pictures, conducted in our laboratory in parallel to the experiments reported here, already showed enhanced SSVEPs for neutral compared to emotional (i.e., unpleasant) IAPS images [70]. We proposed that these unexpected results were due to a superposition effect of an occipito-temporal ERP component called early posterior negativity (EPN), whose amplitude is typically increased in response to emotional as opposed to neutral material [74–76]. The amplitude of the EPN is usually more *positive* for neutral compared to unpleasant images. We surmised that, at a 6-Hz stimulation rate, the EPN components serially elicited by each individual image might superimpose and consequently create larger SSVEPs for neutral RSVP streams. Follow-up studies showed that the increased SSVEP in response to neutral stimuli was limited to 6 Hz and not observed at 3 Hz, 4 Hz, and 8.57 Hz [26], providing additional indirect support that only stimulation frequencies long enough to allow the EPN to fully develop—but not too long to include subsequent ERPs—would elicit such an effect. In another study [77], we presented participants with a task-relevant random dot kinematogram superimposed onto a stream of neutral or

emotional images and flickering at 4 or 6 Hz, respectively. Results showed prioritized processing of background (i.e., task-irrelevant) emotional compared to neutral stimuli at 4 Hz, but a reverse effect at 6 Hz. Another study [27] employed a paradigm very similar to *Experiment 3* but using complex pictures instead of facial expressions. Results showed increased SSVEPs for pleasant compared to neutral and unpleasant pictures in the regularity frequency (i.e., 2 Hz). The difference between neutral and unpleasant conditions was not statistically significant, despite a trend for larger responses to neutral scenes.

Following a similar logic, enhanced SSVEPs in regular neutral conditions reported in the present paper might be a consequence of a smaller negativity (i.e., larger positivity) for neutral compared to angry faces within 200 msec after stimulus onset. In the case of facial stimuli, the corresponding ERP component would be the N170, the first component that more robustly differentiates between facial and non-facial objects [19,21] as well as neutral vs. emotional facial expressions [73,78–80]. According to the dominant view in the literature, however, a serial superposition of the N170 should lead to *greater* regularity responses for angry instead of neutral faces (i.e. our working hypothesis throughout the whole research program). More importantly, this interpretation is inconsistent with the results of *Pilot 1* and *Pilot 2*, where the stimulation and regularity frequencies were much faster (15 Hz and 5 Hz, respectively).

## The role of isolated facial features

In their seminal paper, Diamond and Carey [81] proposed that the visual system uses three types of information to recognize faces: (a) isolated features, such as eyes and mouth; (b) first-order relational/spatial properties (i.e., the eyes are above the nose); and (c) second-order relational information (i.e., the distance between the eyes, or from mouth to nose). More recently, Calvo et al. [82] suggested a functional model of face processing comprising of a serial sequence of steps, with individual features being processed first and very rapidly (< 170 ms) followed by the processing of configurational information, then followed by the analysis of affective cues (around 800 ms after stimulus onset). This is reminiscent of a classical neuroanatomical model of face perception [83] proposing a functional subdivision of a network of occipitotemporal brain areas into two systems: the "core system", responsible for the initial processing of essential facial constituents (e.g., eyes, mouth) in the inferior occipital gyri, the analysis of dynamic features (e.g., eye gaze) in the superior temporal sulcus, and invariant aspects (e.g., unique identity) in lateral fusiform gyrus; and the "extended system", comprising of various brain regions involved in higher-order cognitive functions such as attention (e.g., intraparietal sulcus), emotion extraction (e.g., amygdala), and speech prosody (e.g., auditory cortex). How may these serial models help us interpret the results reported here?

In our experiments, the rapid presentation of individual images within an RSVP stream would presumably allow only for the initial perceptual processes to complete (e.g., interpretation of eyes and mouth and relations among these elements) and little time to analyze more complex information, including emotional cues—especially if we assume strong forward/backward masking effects [3]. This interpretation is further consistent with the absence of a robust inversion effect, i.e. a processing advantage of upright presented faces across all stimulation frequencies. Specifically, inversion might have been ineffective because diagnostic information from each face was extracted from individual features (mainly eyes and mouth), whose processing may be less affected by face inversion [84,85, but see 86 for a different perspective].

Having said that, our participants' brain activity was still differentially modulated by neutral and angry facial expressions. If not emotion, what kind of perceptual information was used by the visual system to diagnostically distinguish between these two conditions? We explored the possibility that within-identity emotion variability could play a role and found some support

for this hypothesis: RSVP streams with low variability elicited a smaller regularity brain response at either 5 Hz or 2 Hz, and differences between neutral and angry faces were either not present (*Pilot 2*) or very small (*Experiment 1* and *Experiment 2*) [see also 66].

A recent study by Poncet et al. [87] used a very similar protocol to the one employed here. These authors presented faces with varying emotional expressions at 6 Hz. Every 5th face (i.e., at 1.2 Hz), either a neutral regular oddball among emotional faces or a regular emotional oddball was presented. Similarly to our findings, their results showed overall larger electrophysiological responses for neutral compared to a range of emotional expressions (along with a selective neural response to fear). Different topographical distributions were also observed among the various emotional faces and neutral ones, suggesting the selective involvement of distinct neuronal networks [in accordance with ref. 83]. Poncet et al. [87] argue that contrasting emotional and non-emotional facial expressions may modulate core regions of the face processing network that differentiate between these two categories, in particular the superior temporal sulcus and the lateral fusiform gyrus which are particularly active in processing changeable features of the faces as well as emotional cues. Such an explanation might also accommodate our findings showing similar SSVEP responses for upright and inverted faces, and more robust results for RSVP streams with highly variable emotional expressions.

Finally, another possible explanation for our findings pertains to the fact that, in this experimental paradigm, neutral faces are the only "non-emotional" stimuli within the RSVP stream, thus serving as "oddball" throughout the whole stimulation. Such an interpretation would assign more weight on theories advocating for general arousal and/or valence mechanisms [88–90]—neutral would be less arousing, and its emotional connotation less prominent, than any other displayed emotion—rather than qualitatively discrete emotional categories [1,91]. This would be in contrast with recent studies interpreting their results as evidence for specific neural processes dedicated to each facial expression [85,90]. The RSVP techniques employed here and elsewhere [28,87,92] are promising tools to disentangle these alternative theoretical interpretations and are a fruitful avenue for future research.

## Conclusions

In a series of experiments, we measured continuous brain responses to rapid serial presentations of faces with emotional expressions and tested whether the visual system prioritizes processing of angry over neutral faces. While manipulating a range of features and applying refinements and additional controls to existing paradigms, our collective results do not support such a processing advantage. Instead, we consistently observed enhanced electrical brain responses during the regular presentation of neutral as opposed to angry faces. We suggest that these findings might be due to the rapid extraction of featural information, particularly within-identity emotion variability: RSVP streams with low variability elicited a smaller regularity brain response, and differences between neutral and angry faces were either not present or very small. Further, we found that hampering face processing by inverting faces had only weak effects on brain responses. Conversely, using facial expressions with high variability yielded stronger brain responses than streams of low-variability expressions. Taken together, our results call for a cautious interpretation of results from experiments using face-stimulus RSVPs to study the processing of emotional expressions—confounds in low-level visual stimulus features might play an important role, and might be more difficult to control than previously acknowledged.

## Supporting information

**S1 File. Supplementary materials.**
(DOCX)

**S1 Fig. Face ratings.** Face ratings for each emotional expression, separately for each participant (single dots), question (*how happy/angry/disgusted is this face*?), and experiment. Mean values are marked by horizontal black lines and 95% confidence intervals represented as transparent boxes. Ratings range from 1 (very low emotional intensity) to 9 (very high emotional intensity).
(TIFF)

**S2 Fig. Reaction times.** Reaction times (in msec) in response to colored dots during the presentation of each face stream, separately for each participant (single dots), condition, and experiment. Mean values are marked by horizontal black lines and 95% confidence intervals represented as transparent boxes.
(TIFF)

**S3 Fig. Cosine similarity at stimulation frequency.** Cosine similarity (CS) calculated at the stimulation frequency for each participant (single dots), condition, and experiment. Mean values are marked by horizontal black lines and 95% confidence intervals represented as transparent boxes.
(TIFF)

**S1 Table. Valence ratings: Descriptives.** Ratings of emotional valence (angry, happy, disgusted) of the different image sets, separately for each experiment, experimental condition, and question.
(XLSX)

**S2 Table. Valence ratings: Analysis.** Statistical comparisons of valence ratings between all pairs of emotional image sets, separately for all experiments and experimental conditions. Mean and 95% HDI are related to differences in rated valence between the respective comparison. Comparisons showing strong evidence against the hypothesis of no difference are presented in bold.
(XLSX)

**S3 Table. Reaction times: Descriptives.** Reaction times to targets presented during trials of the different experimental conditions, separately for each experiment with a behavioral task (i.e., excluding Experiment 2).
(XLSX)

**S4 Table. Reaction times: Analysis.** Statistical comparisons of reaction times between all pairs of factor levels, separately for all experiments and experimental conditions.
(XLSX)

**S5 Table. Cosine similarity at stimulation frequency: Descriptives.** Statistics of Cosine Similarity Index (CS) of the signals at the stimulation frequency, separately for the different experiments and experimental conditions.
(XLSX)

**S6 Table. Cosine similarity at stimulation frequency: Analysis.** Statistical comparisons of Cosine Similarity values between all pairs of factor levels, separately for all experiments and experimental conditions. Mean and 95% HDI are related to differences in reaction times between the respective comparison. Comparisons showing strong evidence against the hypothesis of no difference are presented in bold.
(XLSX)

## Acknowledgments

The experimental stimulation was realized using Cogent Graphics developed by John Romaya at the Laboratory of Neurobiology, Wellcome Department of Imaging Neuroscience, University College London (UCL). We would like to thank Renate Zahn for her help with data collection.

## Author Contributions

**Conceptualization:** Antonio Schettino, Emanuele Porcu, Christian Keitel, Matthias M. Müller.

**Data curation:** Antonio Schettino, Emanuele Porcu, Christopher Gundlach, Christian Keitel.

**Formal analysis:** Antonio Schettino, Emanuele Porcu, Christian Keitel.

**Funding acquisition:** Matthias M. Müller.

**Investigation:** Antonio Schettino, Emanuele Porcu, Christopher Gundlach.

**Methodology:** Antonio Schettino, Emanuele Porcu, Christian Keitel.

**Project administration:** Antonio Schettino, Emanuele Porcu, Christopher Gundlach.

**Resources:** Matthias M. Müller.

**Software:** Antonio Schettino, Emanuele Porcu, Christopher Gundlach, Christian Keitel, Matthias M. Müller.

**Supervision:** Antonio Schettino, Christian Keitel, Matthias M. Müller.

**Validation:** Antonio Schettino, Emanuele Porcu, Christopher Gundlach, Christian Keitel.

**Visualization:** Antonio Schettino, Christopher Gundlach, Christian Keitel.

**Writing – original draft:** Antonio Schettino, Emanuele Porcu, Christopher Gundlach, Christian Keitel, Matthias M. Müller.

**Writing – review & editing:** Antonio Schettino, Emanuele Porcu, Christopher Gundlach, Christian Keitel, Matthias M. Müller.

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
