## [Decision Letter · Decision Letter 0]

10 Jan 2020

PONE-D-19-32180

Rapid processing of neutral and angry expressions within ongoing facial stimulus streams: Is it all about isolated facial features?

PLOS ONE

Dear Dr. Schettino,

Thank you for submitting your manuscript to PLOS ONE. As you can see from their comments both reviewers were generally positive about your manuscript. My own reading of the manuscript is also positive. Therefore, i would like to invite you to resubmit a revised version of your work that carefully considers all these comments. Please, make sure to pay special attention to the reformulation of hypotheses and to the issue regarding the interpretation of your data in terms of task demands.

We would appreciate receiving your revised manuscript by Feb 24 2020 11:59PM. To enhance the reproducibility of your results, we recommend that if applicable you deposit your laboratory protocols in protocols.io, where a protocol can be assigned its own identifier (DOI) such that it can be cited independently in the future. For instructions see: http://journals.plos.org/plosone/s/submission-guidelines#loc-laboratory-protocols

We look forward to receiving your revised manuscript.

Kind regards,

José A Hinojosa, Ph.D.

Academic Editor

PLOS ONE

Journal Requirements:

"The experiments were conducted in accordance with the guidelines of the ethics committee of the University of Leipzig."

a) Please amend your current ethics statement to confirm that your named institutional review board or ethics committee specifically approved this study.

3. We note that Figure(s) 1 and 3 in your submission contain copyrighted images. All PLOS content is published under the Creative Commons Attribution License (CC BY 4.0), which means that the manuscript, images, and Supporting Information files will be freely available online, and any third party is permitted to access, download, copy, distribute, and use these materials in any way, even commercially, with proper attribution. For more information, see our copyright guidelines: http://journals.plos.org/plosone/s/licenses-and-copyright.

1.    You may seek permission from the original copyright holder of Figure(s) 1 and 3 to publish the content specifically under the CC BY 4.0 license.

4. We note that Figures 1 and 3 include an image of a [patient / participant / in the study]. 

Additional Editor Comments (if provided):

Reviewers' comments:

Reviewer's Responses to Questions

**Comments to the Author**

1. Is the manuscript technically sound, and do the data support the conclusions?

Reviewer #1: Yes

Reviewer #2: Yes

2. Has the statistical analysis been performed appropriately and rigorously? 

Reviewer #1: Yes

Reviewer #2: Yes

3. Have the authors made all data underlying the findings in their manuscript fully available?

Reviewer #1: Yes

Reviewer #2: Yes

4. Is the manuscript presented in an intelligible fashion and written in standard English?

Reviewer #1: Yes

Reviewer #2: Yes

5. Review Comments to the Author

Reviewer #1: The studies by Schettino and colleagues investigated the electrophysiological correlates of facial expressions (i.e., neutral vs. angry) processing by means of a fast periodic stimulation (RSVP) approach. The Authors presented four different EEG studies in which several parameters/factors have been manipulated to assess the brain responses underlying the hypothesized processing advantage for emotional (vs. neutral) face expressions. These factors included the stimulus presentation rate (fast-15Hz vs. slow-6Hz), variability across facial expressions (high vs. low), and task (passive observation vs. secondary task). The main result reported was against the central hypothesis, but, more importantly, it was consistent throughout both the pilots and main experiments.

I think that the paper is well-written and technically sound, and it adheres to the standards of the Open Science framework.

Despite that, several aspects should be addressed and improved by the authors that would strengthen their manuscript.

Main comments:

1. Introduction. The introduction is well understandable, but it raises expectations (not respected) in the reader about results in line with the central hypothesis of the Authors. In the discussion section, the Authors reported several findings that are consistent with their findings (enhanced SSVEPs in response to neutral vs. emotional stimuli). In my humble opinion, the Authors could reorganize the Introduction by introducing and anticipating both hypotheses: the main one (enhanced SSVEPs to angry stimuli) and the possibility for the opposite one (supported by the literature in the General Discussion). The Bayesian approach, already used, could be suitable for testing the direction of such effect. Given that the Authors proposed four different studies to assess the role of different factors in modulating face processing, such a rigorous approach should be highlighted from the beginning. This would emphasize the consistency in the results and be beneficial to the entire manuscript.

2. Methods: Spectral decomposition of stimulus-driven EEG signals.

The authors employed spatial filtering (RESS) introduced by Cohen and Gulbinaite (2017). I wonder if a more “traditional” approach based on a single or group (ROI) electrode selection would lead to different results. Either a convergence or comparison between the two techniques of analysis would strengthen the manuscript. In this regard, what about possible hemispheric asymmetries previously reported during face (also emotional expression) processing? Did the Authors expect, or check, for any gender-related difference in brain responses, since their groups of participants included both male and female individuals?

3. Discussions and General Discussion. I think that each Discussion section would benefit from the inclusion of a few bibliographical references in support of the main results. This would help to link each section to the following experiment proposed.

In the General Discussion, I truly appreciated the statement about possible post-hoc interpretations of the results made by the Authors. However, under my comment about reorganizing the Introduction, I would suggest anticipating previous evidence reported here (consistent with that of the Authors) by reporting it in the earlier sections of the manuscript (i.e., Introduction and Interim Discussions). In this regard, the last paragraphs are dense with data and notions and seem to be written with a different style. Why not use previous studies to support, instead of justifying, the (stable) results from the present work?

Minor comments:

1. In both the introduction and General Discussion, the Authors referred to several studies without reporting the name of the principal author. This style of writing simplifies the text but could make it difficult to understand if the Authors are referring to their previous works or to those of other research groups. I suggest increasing clarity on this point.

2. In the plots regarding CS values (i.e., Figures 2, 4, 5, 6), it would be nice to mark (i.e., “*”) the significant differences between conditions that have been found. This would help to guide the reader through the results in the different studies. This suggestion also applies to supplementary figures reporting subjective ratings.

Reviewer #2: The series of experiments performed by Schettino and colleagues sought to investigate the degree to which the visual system can extract emotional information from faces during an RSVP paradigm. Specifically, the question was to determine what the ideal 'extraction speed' would be, hypothesizing that angry faces would be preferentially processed and therefore lead to an enhanced ssVEP over neutral. Surprisingly, neutral faces consistently, over 4 experiments, showed enhanced processing via the ssVEP.

The authors nicely explain this neutral face effect with various hypotheses in their discussion. Both hypotheses (superposition effect and isolated facial features) are viable. However, a third option could be that across identities, the expression of 'angry' is variable, whereas neutral is highly consistent. Although testing for within-identity variability in facial expression (how was dissimiliarity of facial expression determined?), the consideration of between-subjects variability for each emotion is worth debating. This would naturally apply more for the first pilot study using multiple identities, but may generally be decoded by the face network system as a reliable, consistent stimulus, that in a stream of highly variable expressions (constantly switching between 3 emotions) evokes a strong signal due to the more easily predictive nature of a consistent neutral face.

Overall, the study with its four experiments were well designed and described. It is reassuring to see topographies remain stable across experiments. The irregular control condition was particular nice, although authors should be careful in saying that "regularity frequencies were compared", as for the irregular condition there was no regularity frequency by design. Minor semantic comment. I would be interested to see the overall ssVEP signal to the RSVP for the irregular condition, however. Despite showing that the regularity frequency was not a subharmonic of the main driving frequency, would be useful to see if this regularity frequency modified the overall RSVP signal in any way, in a multiplicative or even destructive fashion. With growing conversation in the literature on superposition, the amplification of fundamental frequencies by non-physical harmonics, so to speak, could potentially add to this literature.

The interpretation of the emotional differences between experiments 1 and 2 should be described with more caution, as task demand interpretation could go either way (enhanced attention, withdrawn attention) and was not primarily tested for in this study.

I would recommend this paper for acceptance with minor revisions. Most of the revisions should be directed at a more streamlined and concise conceptual setup in the introduction. For example the reader begins by thinking the point of the paper is timing of emotional extraction, but then aside from two fundamental frequencies (15 and 6Hz), timing/frequency is not parametrically modulated. Instead, the story evolves into wanting to show that threat should evoke a greater response than neutral. The introduction reads more like an extended methods section than a conceptual walk through, with not enough ‘why we should care’. Overall, less conversational language should be used, which is particularly pronounced in the discussion. This work is timely and contributes to face processing literature as well as expanding our mechanistic understanding of ssVEP signals and their caveats.

6. PLOS authors have the option to publish the peer review history of their article (what does this mean?). If published, this will include your full peer review and any attached files.

Reviewer #1: No

Reviewer #2: No

---

## [Author Response · Author response to Decision Letter 0]

28 Feb 2020

Reviewers' comments

Reviewer #1

The studies by Schettino and colleagues investigated the electrophysiological correlates of facial expressions (i.e., neutral vs. angry) processing by means of a fast periodic stimulation (RSVP) approach. The Authors presented four different EEG studies in which several parameters/factors have been manipulated to assess the brain responses underlying the hypothesized processing advantage for emotional (vs. neutral) face expressions. These factors included the stimulus presentation rate (fast-15Hz vs. slow-6Hz), variability across facial expressions (high vs. low), and task (passive observation vs. secondary task). The main result reported was against the central hypothesis, but, more importantly, it was consistent throughout both the pilots and main experiments.

I think that the paper is well-written and technically sound, and it adheres to the standards of the Open Science framework.

We thank the Reviewer for the positive evaluation and constructive feedback.

Despite that, several aspects should be addressed and improved by the authors that would strengthen their manuscript.

Main comments:

1. Introduction. The introduction is well understandable, but it raises expectations (not respected) in the reader about results in line with the central hypothesis of the Authors. In the discussion section, the Authors reported several findings that are consistent with their findings (enhanced SSVEPs in response to neutral vs. emotional stimuli). In my humble opinion, the Authors could reorganize the Introduction by introducing and anticipating both hypotheses: the main one (enhanced SSVEPs to angry stimuli) and the possibility for the opposite one (supported by the literature in the General Discussion). The Bayesian approach, already used, could be suitable for testing the direction of such effect. Given that the Authors proposed four different studies to assess the role of different factors in modulating face processing, such a rigorous approach should be highlighted from the beginning. This would emphasize the consistency in the results and be beneficial to the entire manuscript.

Following the Reviewer’s suggestions, we integrated our Introduction and Abstract sections to more clearly emphasize our initial starting hypothesis: processing advantages for angry facial expressions should manifest in an increased SSVEP amplitude compared to neutral faces. The opposite pattern (attenuated SSVEPs for threatening stimuli) was initially unanticipated. In the literature, initial support for this opposite pattern only accrued in the course of this research program and was thus not previously available (also, to our knowledge, the majority of findings in the current literature on this topic still report larger SSVEP amplitude for emotion-laden as opposed to neutral information). We believe that, in its current form, our manuscript faithfully reflects this timeline. We prefer to keep this structure to avoid potential misinterpretations, i.e., finding support for a hypothesis we did not formulate a priori.

2. Methods: Spectral decomposition of stimulus-driven EEG signals.

The authors employed spatial filtering (RESS) introduced by Cohen and Gulbinaite (2017). I wonder if a more “traditional” approach based on a single or group (ROI) electrode selection would lead to different results. Either a convergence or comparison between the two techniques of analysis would strengthen the manuscript. In this regard, what about possible hemispheric asymmetries previously reported during face (also emotional expression) processing? Did the Authors expect, or check, for any gender-related difference in brain responses, since their groups of participants included both male and female individuals?

While we appreciate the reasoning behind the Reviewer’s request to provide more “traditional” analyses, we would argue against their inclusion in this manuscript, for several reasons. First, re-running all analyses and reporting their results would inflate an already voluminous paper, hence diluting the central findings. Second, Cohen and Gulbinaite (2017) already provide an in-principle comparison between the traditional and the RESS approach. Please note that we are not advocating the use of RESS as a superior analysis method for SSVEPs in general. However, in our series of experiments, we expected (and found) topographic differences in regularity-driven SSVEP between experimental conditions; therefore, defining amplitude-based ROIs would be suboptimal (we clarified this point in the revised manuscript, page 17). A traditional, ROI-based approach might produce slightly different findings -- although, we surmise, with no qualitative differences in the pattern of results reported in our manuscript. However, we would be inclined to give the RESS-derived findings more weight because they do not rest on assumptions imposed by ROI/electrode selections. Instead, RESS better adapts to topographic differences between experimental conditions: see, for instance, the scalp maps for upright vs. inverted faces in Fig. 2 or low vs. high expression variability in Figs. 4-5-6. The interested reader is encouraged to reuse and reanalyze our data (available at https://osf.io/uhczc/ under a Creative Commons Attribution 4.0 International Public License) with any approach they are interested in.

In a similar vein, we would also refer to the availability of the data for any follow-up exploratory analysis of hemispheric or gender differences. We believe it would not be appropriate to include them in the present manuscript not only for reasons of succinctness, but also because such exploratory questions were outside the scope of our investigation.

3. Discussions and General Discussion. I think that each Discussion section would benefit from the inclusion of a few bibliographical references in support of the main results. This would help to link each section to the following experiment proposed.

In the General Discussion, I truly appreciated the statement about possible post-hoc interpretations of the results made by the Authors. However, under my comment about reorganizing the Introduction, I would suggest anticipating previous evidence reported here (consistent with that of the Authors) by reporting it in the earlier sections of the manuscript (i.e., Introduction and Interim Discussions). In this regard, the last paragraphs are dense with data and notions and seem to be written with a different style. Why not use previous studies to support, instead of justifying, the (stable) results from the present work?

In line with the Reviewer’s comment, we have added supporting references to the Discussion sections of each experiment. Regarding the General Discussion, as explained above, we prefer to keep the current structure because it highlights the timeline of our research programme. Some of the cited studies, pivotal for subsequent design decisions and current interpretation of the results, were not published at the time we conceived and began the research program.

Minor comments:

1. In both the introduction and General Discussion, the Authors referred to several studies without reporting the name of the principal author. This style of writing simplifies the text but could make it difficult to understand if the Authors are referring to their previous works or to those of other research groups. I suggest increasing clarity on this point.

We thank the Reviewer for pointing this out. We now clarify the corresponding parts accordingly throughout the text.

2. In the plots regarding CS values (i.e., Figures 2, 4, 5, 6), it would be nice to mark (i.e., “*”) the significant differences between conditions that have been found. This would help to guide the reader through the results in the different studies. This suggestion also applies to supplementary figures reporting subjective ratings.

We would like to emphasize that, from a statistical perspective, “significance” is not defined in the present context. We provide rule-of-thumb thresholds for Evidence Ratios (ERs) for ease of readability; however, we encourage readers to appreciate the rich evidential value of continuous measures of Bayesian inference (for an introduction, see Kruschke & Liddell, 2017). For this reason, it would be inconsistent to mark comparisons in the Figures as suggested by the Reviewer. Furthermore, the large number of comparisons would result in many symbols in each figure, consequently impairing their readability.

The Figures are meant to illustrate the distribution of the data, for example by clearly showing no discernible regularity activity in the irregular conditions as well as differences between low and high within-identity emotion variability in upright faces (see Fig 2). Nonetheless, we appreciate the Reviewer’s suggestion to better guide readers through the results of the different studies. Therefore, we now ensure that, in each figure caption, the corresponding table with relevant statistical results are appropriately referenced.

Reviewer #2

The series of experiments performed by Schettino and colleagues sought to investigate the degree to which the visual system can extract emotional information from faces during an RSVP paradigm. Specifically, the question was to determine what the ideal 'extraction speed' would be, hypothesizing that angry faces would be preferentially processed and therefore lead to an enhanced ssVEP over neutral. Surprisingly, neutral faces consistently, over 4 experiments, showed enhanced processing via the ssVEP.

We thank the Reviewer for the careful consideration of our manuscript as well as the positive remarks throughout the review.

The authors nicely explain this neutral face effect with various hypotheses in their discussion. Both hypotheses (superposition effect and isolated facial features) are viable. However, a third option could be that across identities, the expression of 'angry' is variable, whereas neutral is highly consistent. Although testing for within-identity variability in facial expression (how was dissimiliarity of facial expression determined?), the consideration of between-subjects variability for each emotion is worth debating. This would naturally apply more for the first pilot study using multiple identities, but may generally be decoded by the face network system as a reliable, consistent stimulus, that in a stream of highly variable expressions (constantly switching between 3 emotions) evokes a strong signal due to the more easily predictive nature of a consistent neutral face.

We agree with the Reviewer. As noted on page. 23: “We speculated that the high variability of the stimulus material -- nine different facial identities displaying three emotional expressions to various degrees of intensity -- might have been a confounding factor. Specifically, angry expressions may differ more between identities than neutral expressions, and this greater dissimilarity may have led to a less consistent brain response.”. This prompted us to separately display facial identities with low and high within-identity emotion variability and assess the contribution of this factor in modulating the SSVEP signal. Another study reporting SSVEP adaptation invariant to changes in facial expression only when the same identity was presented (Vakli et al., 2014) is now cited (e.g., on page 24).

We took this opportunity to respond to another request of the Reviewer, i.e., clarify how we determined within-identity emotion variability in our experiments (see page 25).

One more point raised by the reviewer merits attention. It is plausible to hypothesize that regular neutral faces received prioritized attention allocation -- and, therefore, elicited larger SSVEP amplitude -- because they were the only non-emotional stimuli in the RSVP stream. We already briefly mentioned this alternative explanation on page 40: “[...] neutral faces might have been processed more intensely because the visual system expected emotional content given the RSVP context (i.e., mostly emotional expressions)”. We now emphasize this interesting alternative explanation in the General Discussion section, page 46-47.

Overall, the study with its four experiments were well designed and described. It is reassuring to see topographies remain stable across experiments. The irregular control condition was particular nice, although authors should be careful in saying that "regularity frequencies were compared", as for the irregular condition there was no regularity frequency by design. Minor semantic comment.

We have clarified the wording in line with the Reviewer’s comment. Specifically, on page 20, we now write: “Throughout the main text we report the results of the analysis carried out on the cosine-similarity values at the Fourier coefficients that correspond to the regularity frequency in respective stimulation conditions (also for irregular presentation conditions).”. 

I would be interested to see the overall ssVEP signal to the RSVP for the irregular condition, however. Despite showing that the regularity frequency was not a subharmonic of the main driving frequency, would be useful to see if this regularity frequency modified the overall RSVP signal in any way, in a multiplicative or even destructive fashion. With growing conversation in the literature on superposition, the amplification of fundamental frequencies by non-physical harmonics, so to speak, could potentially add to this literature.

The Reviewer is invited to browse the Supplementary Materials included with our submission -- specifically, the Stimulation Frequency section --, where we report a full analysis of 15Hz (pilots) and 6Hz (experiments) SSVEPs driven by the RSVP. In brief, we observed increased attention allocation during highly variable (upright) face streams, particularly for regular angry and irregular conditions. Thus, there does not seem to be robust evidence that regularity specifically affected (in a multiplicative or destructive way) the SSVEP signal at the stimulation frequency. However, as highlighted in the Supplementary Materials, we advise caution in the interpretation of these findings given their heterogeneous pattern, and hope that future studies will shed light on this issue more systematically.

The interpretation of the emotional differences between experiments 1 and 2 should be described with more caution, as task demand interpretation could go either way (enhanced attention, withdrawn attention) and was not primarily tested for in this study.

We thank the Reviewer for pointing out this caveat. This information has now been added on page 41.

I would recommend this paper for acceptance with minor revisions. Most of the revisions should be directed at a more streamlined and concise conceptual setup in the introduction. For example the reader begins by thinking the point of the paper is timing of emotional extraction, but then aside from two fundamental frequencies (15 and 6Hz), timing/frequency is not parametrically modulated. Instead, the story evolves into wanting to show that threat should evoke a greater response than neutral. The introduction reads more like an extended methods section than a conceptual walk through, with not enough ‘why we should care’. Overall, less conversational language should be used, which is particularly pronounced in the discussion. This work is timely and contributes to face processing literature as well as expanding our mechanistic understanding of ssVEP signals and their caveats.

We thank the Reviewer for this observation, which mirrors Reviewer #1’s comments. We have now clarified and emphasized the main hypotheses in the Abstract and Introduction sections.

---

## [Decision Letter · Decision Letter 1]

6 Apr 2020

Rapid processing of neutral and angry expressions within ongoing facial stimulus streams: Is it all about isolated facial features?

PONE-D-19-32180R1

Dear Dr. Schettino,

We are pleased to inform you that your manuscript has been judged scientifically suitable for publication and will be formally accepted for publication once it complies with all outstanding technical requirements.

With kind regards,

José A Hinojosa, Ph.D.

Academic Editor

PLOS ONE

Additional Editor Comments (optional):

Reviewers' comments:

Reviewer's Responses to Questions

**Comments to the Author**

1. If the authors have adequately addressed your comments raised in a previous round of review and you feel that this manuscript is now acceptable for publication, you may indicate that here to bypass the “Comments to the Author” section, enter your conflict of interest statement in the “Confidential to Editor” section, and submit your "Accept" recommendation.

Reviewer #1: All comments have been addressed

Reviewer #2: All comments have been addressed

2. Is the manuscript technically sound, and do the data support the conclusions?

Reviewer #1: Yes

Reviewer #2: Yes

3. Has the statistical analysis been performed appropriately and rigorously? 

Reviewer #1: Yes

Reviewer #2: Yes

4. Have the authors made all data underlying the findings in their manuscript fully available?

Reviewer #1: Yes

Reviewer #2: Yes

5. Is the manuscript presented in an intelligible fashion and written in standard English?

Reviewer #1: Yes

Reviewer #2: Yes

6. Review Comments to the Author

Reviewer #1: Consistent with reviewers' suggestions, the authors have made changes in their manuscript that increased the overall readability and understanding of the work. The hypotheses are better framed in the introduction, which led to enhanced consistency through the following sections.

Reviewer #2: The authors have nicely addressed all concerns that I raised and I am happy to see this work published. I particularly applaud the authors on the rigor of their 4-experimental setup.

7. PLOS authors have the option to publish the peer review history of their article (what does this mean?). If published, this will include your full peer review and any attached files.

Reviewer #1: No

Reviewer #2: No

---

## [Editor Report · Acceptance letter]

9 Apr 2020

PONE-D-19-32180R1 

Rapid processing of neutral and angry expressions within ongoing facial stimulus streams: Is it all about isolated facial features? 

Dear Dr. Schettino:

I am pleased to inform you that your manuscript has been deemed suitable for publication in PLOS ONE. Congratulations! Your manuscript is now with our production department. 

With kind regards,

on behalf of

Dr. José A Hinojosa 

Academic Editor

PLOS ONE